# Sexual dimorphism in sensorimotor transformation of optic flow

**Sarah Nicholas[1], Katja Sporar Klinge[1†], Luke Turnbull[1], Annabel Moran[1], Aika Young[1], Yuri Ogawa[1]\*, Karin Nordström[1,2]\***

[1]Flinders Health and Medical Research Institute, Flinders University, Adelaide, Australia; [2]Department of Medical Cell Biology, Uppsala University, Uppsala, Sweden

**\*For correspondence:**
yuri.ogawa@flinders.edu.au (YO);
karin.nordstrom@flinders.edu.au (KN)

**Present address:** [†]Institute for Developmental Biology, RWTH Aachen University, Aachen, Germany

**Competing interest:** The authors declare that no competing interests exist.

## eLife Assessment

Hoverflies are known for their sexually dimorphic visual systems and exquisite flight behaviors. This **valuable** study reports how two types of visual descending neurons differ between males and females in their motion- and speed-dependent responses, yet surprisingly, the behavior they control lacks any sexual dimorphism. The results **convincingly** support these findings, which will be of interest for studies of visuomotor transformations and network-level brain organization.

**Abstract** Motion vision underpins a wide range of adaptive behaviours essential for individual and species survival. In hoverflies, some visual behaviours are sexually dimorphic, including for example male high-speed pursuit of conspecifics matched by improved optics and faster photoreceptors. Other visual behaviours are sexually monomorphic, with for example similar foraging flight speeds in male and female hoverflies. However, whether the descending neurons responsible for sensorimotor transformation of optic flow are sexually dimorphic is unknown. To address this, we combined morphological analysis with electrophysiology of optic flow sensitive descending neurons and compared neural responses to the behavioural output in tethered hoverflies. We found that while optomotor flight behaviour is largely sexually monomorphic, the underlying neural responses are sexually dimorphic, especially at higher optic flow velocities. Additionally, behavioural responses were noticeably slower than neural responses. Together, our findings uncover a nuanced, sex- and stimulus-dependent sensorimotor transformation, shaped by both neural architecture and behavioural demands.

## Introduction

Motion vision is a fundamental sensory modality across the animal kingdom, enabling animals to navigate, to maintain a straight trajectory, to avoid collisions, and to identify prey, predators, or mates. Among the most potent cues for self-motion is widefield optic flow, the coherent motion of the entire visual field, generated by an animal's own movement through the world. In insects, the neural mechanisms underlying optic flow processing have been studied extensively (for a review, see e.g. *Mauss and Borst, 2020*), offering a powerful model for understanding how compact nervous systems extract behaviourally relevant information from dynamic visual scenes.

To generate appropriate responses to widefield optic flow, the visual input needs to be integrated across space. In flies, this spatial pooling occurs in 45–60 lobula plate tangential cells (LPTCs) (*Pierantoni, 1976*; *Zhao et al., 2023*), each matched to a particular type of self-generated optic flow (*Franz and Krapp, 2000*), where the horizontal system (HS) and vertical system (VS) cells are the most well described. HS cells respond optimally to rotations around the yaw axis, whereas VS cells are tuned to pitch and roll rotations (*Krapp et al., 1998*). LPTCs project to the inferior posterior slope (*Wei*

*et al., 2020*), where they synapse with descending neurons (*Strausfeld and Bassemir, 1985a*; *Strausfeld and Seyan, 1985b*). In *Drosophila* at least 35 descending neuron types have their inputs in the posterior surface of the brain (named DNp1-35) (*Namiki et al., 2018*). Furthermore, in *Drosophila* and blowflies, three of these descending neurons have been shown to respond robustly to widefield optic flow. Their axons project to the dorsal part of the thoracic ganglion, where motor neurons controlling the neck, wings and halteres are located (*Strausfeld and Bassemir, 1985a*; *Suver et al., 2016*; *Erginkaya et al., 2023*; *Haikala et al., 2013*; *Pokusaeva et al., 2024*).

DNp15, also called DNHS1, receives input from HS cells (*Namiki et al., 2018*; *Suver et al., 2016*; *Erginkaya et al., 2023*) and is physiologically similar to the optic flow sensitive descending neuron type 1 (OFS DN1) in the hoverfly *Eristalis tenax* (*Nicholas et al., 2020a*), although direct LPTC coupling has not yet been demonstrated in the hoverfly. DNHS1 projects to the neck and haltere motor neuropils (*Namiki et al., 2018*) and has been implicated in the control of head yaw movements, abdominal ruddering and flight stabilization via the haltere motor system (*Suver et al., 2016*). DNp20, or DNOVS1, receiving input from the ocelli and VS cells (*Strausfeld and Bassemir, 1985a*; *Namiki et al., 2018*; *Suver et al., 2016*; *Haag et al., 2007*), is likely involved in rapid head movements (*Namiki et al., 2018*), possibly facilitating gaze stabilization during flight (*Suver et al., 2016*). DNp22, or DNOVS2, also receives input from the ocelli but a different subset of VS cells (*Strausfeld and Bassemir, 1985a*; *Namiki et al., 2018*; *Suver et al., 2016*; *Wertz et al., 2008*). DNOVS2 is physiologically similar to the *E. tenax* optic flow sensitive descending neuron type 2 (OFS DN2) (*Nicholas et al., 2020a*), but synaptic coupling with VS cells has not been shown in hoverflies. DNOVS2 projects to the neck, wing and haltere motor neuropils (*Namiki et al., 2018*) and has been implied in the initiation of the fast body saccades that support rapid re-orientation (*Suver et al., 2016*) during flight in response to dynamic visual cues.

Hoverflies are interesting in the context of motion vision, which they use both to maintain a hovering stance and to fly at high speed (*Collett and Land, 1975*). Indeed, hoverflies show striking sexually dimorphic flight behaviour, where males establish territories which they guard rigorously from a hovering stance (*Wellington and Fitzpatrick, 1981*), followed by high-speed flight to chase away any intruding insects and/or pursue conspecific females for courtship and mating (*Collett and Land, 1975*; *Collett and Land, 1978*). Male hoverflies are also smaller than females (*Daňková et al., 2023*; *Nicholas et al., 2018b*), which introduces an aerodynamic component by reducing inertia and enabling finer control over rapid flight adjustments (*Dudley, 2002*). Accompanying this sexually dimorphic pursuit behaviour, male *E. tenax* have larger lenses than females in a dorso-frontal bright zone (*Straw et al., 2006*), with faster motion detection and increased signal-to-noise ratio (*van Hateren et al., 1989*). In many fly species, the photoreceptors in this part of the eye are also faster in males (*Hornstein et al., 2000*). In hoverflies, even if the LPTCs are typically implied in optomotor responses (*Mauss and Borst, 2020*), males have a smaller HSN receptive field (*Nordström et al., 2008*) and velocity tuning shifted to higher velocities (*Straw et al., 2006*; *Barnett et al., 2010*). These adaptations are likely useful in regulating optomotor responses during high-speed target pursuit (*Nicholas and Nordström, 2021*; *Ghosh et al., 2025*). Indeed, as target selective descending neurons (TSDNs) are suppressed when target and background move in the same direction (*Nicholas and Nordström, 2021*; *Nicholas et al., 2018a*), the optic flow sensitive descending neurons may serve a complementary role in stabilizing flight under these conditions.

The differences in optics, photoreceptor dynamics, and LPTC receptive field size and velocity tuning have been interpreted as required by males in the fast flight used during sexually dimorphic territorial behaviours. Interestingly, there is no sexual dimorphism in flight speed during other behaviours likely to be governed primarily by the LPTCs, such as foraging between flowers (*Thyselius et al., 2018*) and when flying within the confines of an indoor arena (*Thyselius et al., 2023*). To investigate the discrepancy between sexually dimorphic visual processing and sexually monomorphic flight speed, we compared the electrophysiological response characteristics and morphology of optic flow sensitive descending neurons in male and female hoverflies and compared these findings to the behaviour of tethered hoverflies viewing similar stimuli. We used the wing beat amplitude (WBA) as a measure of the optomotor response and found no sexual dimorphism at speeds up to 2 m/s for translation and 200°/s for rotation. While the head movements were largely sexually monomorphic, the extension of the fore- and hind legs exhibited clearer sexual dimorphism. Furthermore, while neural morphology, receptive fields and direction sensitivity of the descending neurons showed minimal sex differences,

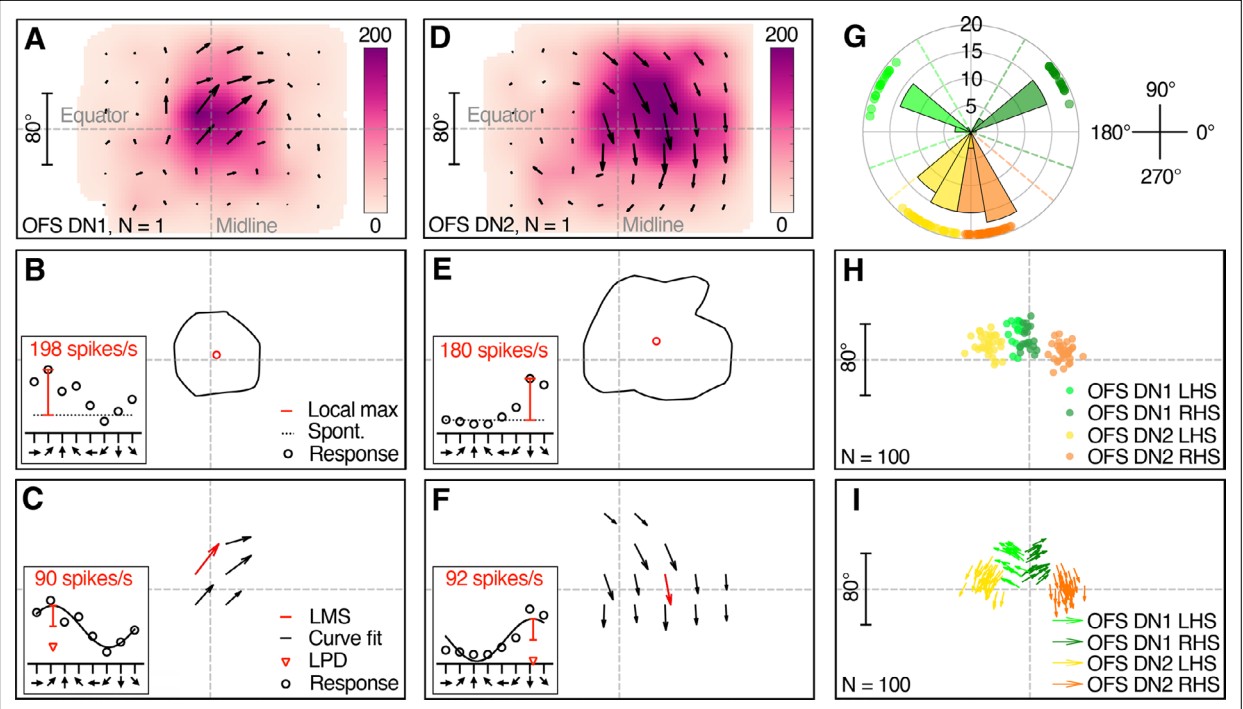

**Figure 1.** Classification of optic flow sensitive (OFS) descending neurons (DNs) in *Eristalis tenax*. (**A**) Receptive field of a representative OFS DN1 recorded from a male hoverfly. Colour coding indicates the local maximum spike frequency (red, inset, panel **B**), the direction of the arrows shows the local preferred direction (LPD, red arrowhead, panel **C**) and their length the local motion sensitivity (LMS, red, panel **C**). As the stimuli were not perspective corrected, the receptive field map reflects the location on the visual monitor. (**B**) Contour line representing the 50% receptive field boundary based on local maximum spike frequency (colour coding, panel **A**). Inset: example response at one central location to eight directions of motion (black circles), showing local maximum spike frequency (local max, red line) above spontaneous rate (spont, black dotted line). (**C**) LPD map of the same example neuron at locations where LMS exceeds 50% of the maximum. Inset: example response at one location to eight directions of motion with a sinusoidal fit (black line), illustrating LMS (red line) and LPD (red arrowhead). The data in the inset corresponds to the red arrow. (**D**) Receptive field of a representative male OFS DN2. (**E**) 50% receptive field contour and receptive field centre of the neuron shown in panel D. (**F**) Preferred direction map of the same OFS DN2. (**G**) Distribution of receptive field preferred directions across 100 male reference neurons, colour-coded by neuron classification. Dashed lines in corresponding colours indicate the thresholds used for neuron type classification. (**H**) Receptive field centres of the same 100 reference neurons, using the same colour coding. (**I**) Relationship between preferred direction and receptive field centre for the 100 neurons. Note that the scale is the same for the azimuth and elevation, and the scale bar in panels **A, D, H, I** corresponds to both the x and y axes.

The online version of this article includes the following figure supplement(s) for figure 1:

**Figure supplement 1.** Identification of four OFS descending neuron clusters based on receptive field properties.

**Figure supplement 2.** Exclusion criteria and receptive field comparisons in male and female OFS DNs.

there was a significant and noticeable difference in the velocity response functions between males and females, especially at higher speeds. Critically, neural differences were not only velocity dependent but also varied between stimuli (occurring for sideslip, lift, and thrust but not roll) and neuron type. These neuron-, stimulus-, and sex-specific differences uncover a previously unrecognized complexity in the neural encoding of visual motion, revealing a sex-dependent transformation from sensory input to motor output.

## Results

### Two distinct types of optic flow sensitive descending neurons can be identified by their receptive field location and preferred direction

Optic flow sensitive descending neurons can be readily identified by mapping their receptive field using small sinusoidal gratings (*Nicholas et al., 2020a*; *Straw et al., 2006*). Based on the receptive fields of 100 reference neurons recorded from 90 male hoverflies, we found that two key parameters, the azimuthal position of the receptive field centre and its preferred direction of motion, are sufficient

to reliably ascertain neuron type (*Figure 1*, *Figure 1—figure supplement 1*). OFS DN1 has a preferred direction up and away from the midline, either leftward (range from 137° to 171°) or rightward (range from 16° to 40°) for neurons on the left- and right-hand side of the visual field, respectively (*Figure 1A–C*; green, *Figure 1G–I*). OFS DN2 responds preferentially to downward motion (range from 228° to 293°; *Figure 1D–F*; yellow and orange, *Figure 1G–I*) with the azimuthal position of the receptive field centre separating left-hand side (LHS) from right-hand side (RHS) neurons (*Figure 1G–I*).

We used the maximum local motion sensitivity (LMS, *Figure 1C, F*), the extent of the receptive fields (number of positions with LMS over 50%, arrows in *Figure 1C, F*) and the local preferred direction (LPD, *Figure 1C, F*) variance from these 100 reference neurons (grey data, *Figure 1—figure supplement 2A–C*) to set strict exclusion criteria (dashed red, *Figure 1—figure supplement 2A–C*) of the neurons used in the rest of the paper. This resulted in the exclusion of two neurons from males and seven from females (grey, *Figure 1—figure supplement 2D, E*) due to either low LMS (less than 20 spikes/s, *Figure 1—figure supplement 2A*), a small number of locations where LMS exceeded 50% of the maximum (four positions or less, *Figure 1—figure supplement 2B*) or high LPD variance (above 30°, *Figure 1—figure supplement 2C*).

We found no sexual dimorphism in either neuron type when comparing receptive field width and height of the remaining 33 male and 29 female neurons (*Figure 1—figure supplement 2F*, unpaired *t*-test, p = 0.52 and 0.09 for width and height of OFS DN1, and p = 0.19 and 0.13 for width and height of OFS DN2).

## Directional tuning of optic flow sensitive descending neurons exhibits limited sexual dimorphism

Some LPTCs, which are upstream of optic flow sensitive descending neurons, show distinct sexual dimorphism, while others do not (*Straw et al., 2006*; *Nordström et al., 2008*). The receptive field data used for classifying OFS DN1 and DN2 showed that they are strongly directional (*Figure 1G, I*). To investigate if this direction tuning is sexually dimorphic, we used a separate experiment quantifying responses to full-screen sinusoidal grating stimuli (wavelength 7°, 5 Hz, *Figure 2—figure supplement 1*). We found that the resulting preferred direction of both OFS DN1 and OFS DN2 matched their receptive

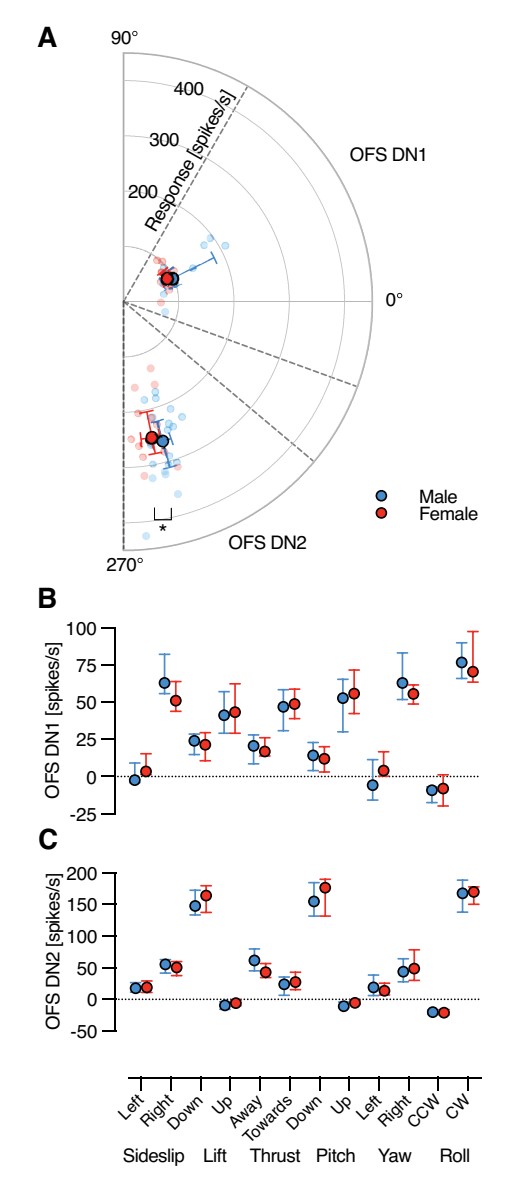

**Figure 2.** Sex-based comparison of direction sensitivity in OFS DNs. (**A**) Polar plot showing the preferred direction of male (blue) and female (red) OFS DNs in response to a full-screen, full-contrast sinusoidal grating (spatial wavelength 7°, temporal frequency 5 Hz) moving in eight different directions (see also *Figure 2—figure supplement 2*). Individual data points represent the response amplitude and preferred direction of each OFS DN (red, *Figure 2—figure supplement 2J*). Larger, salient circles indicate the population median, with error bars showing the interquartile range. The dashed lines indicate the directional thresholds used to classify neuron type, as OFS DN1 (male: *N* = 9; female: *N* = 12) or OFS DN2 (male: *N* = 20; female: *N* = 14). Asterisk indicates a statistically significant difference (p < 0.05, Watson–Williams two-sample test). (**B**) Comparison of male and

*Figure 2 continued on next page*

*Figure 2 continued*

female OFS DN1 responses to translational optic flow at 0.5 m/s: sideslip, lift, and thrust; and rotational optic flow at 50°/s: pitch, yaw, and roll (male: N = 9; female: N = 12). (**C**) Comparison of male and female OFS DN2 responses to the same optic flow stimuli (male: N = 20; female: N = 14). Data presented as median and interquartile range.

The online version of this article includes the following figure supplement(s) for figure 2:

**Figure supplement 1.** Schematic illustration showing the steps used to standardize data across recordings.

**Figure supplement 2.** Comparison of spontaneous activity and responses to stationary stimuli.

field preferred directions (compare polar plots in *Figure 1—figure supplement 2D, E*, with *Figure 2—figure supplement 1G, H*), that is up and away from the visual midline for OFS DN1 (range from 359° to 52°) or downwards for OFS DN2 (range from 273° to 297°, *Figure 2A*). While OFS DN1 showed no difference in preferred direction between the sexes (Watson–Williams two-sample test, p = 0.38), male OFS DN2 had a slightly more lateral preferred direction compared to females (*Figure 2A*; median = 285.4° compared to 281.5°, Watson–Williams two-sample test, p = 0.046).

We quantified the spontaneous rate and found that neither this nor the responses of OFS DN1 to a stationary starfield pattern differed between the sexes (circles, *Figure 2—figure supplement 2A*, two-way ANOVA, p = 0.29). Conversely, OFS DN2 exhibited significant sexual dimorphism, with females displaying a higher spontaneous rate and response to stationary stimuli than males (circles, *Figure 2—figure supplement 2B*, two-way ANOVA, p < 0.001). The response to a stationary stimulus was larger than spontaneous rate in males (p = 0.015), but not in females (red circles, *Figure 2—figure supplement 2B*, p = 0.35, two-way ANOVA).

We next looked at the responses to moving full-screen three-dimensional starfield stimuli simulating the type of optic flow that would be generated by self-motion through space (*Leibbrandt et al., 2021*) (at 0.5 m/s for translations and 50°/s for rotations), after subtracting the response to the stationary stimulus (filled circles, *Figure 2—figure supplement 2A, B*). As predicted from the receptive fields (*Figure 1*) OFS DN1 was excited by stimuli moving either up or away from the midline, such as rightward sideslip and yaw, and upwards lift and pitch (*Figure 2B*), while OFS DN2 showed the strongest responses to downwards lift and pitch (*Figure 2C*). Both neuron types respond strongly to clockwise roll (*Figure 2B, C*), as predicted by their receptive fields (*Figure 1A, B*) and shown previously for males (*Nicholas et al., 2020a*). However, neither neuron type showed any sexual dimorphism (*Figure 2B, C*, two-way ANOVA, p = 0.81 and 0.92, OFS DN1 and OFS DN2, respectively).

## Sexual dimorphism in optic flow descending neurons is velocity dependent

As the velocity tuning of some LPTCs is sexually dimorphic (*Straw et al., 2006*; *Barnett et al., 2010*), we tested the responses of optic flow sensitive descending neurons using a continuous velocity step stimulus (*Figure 3*, *Video 1*). Six velocities for each direction of the stimuli (e.g. anticlockwise roll –10 to –200°/s and clockwise roll +10 to +200°/s) and a stationary control were presented three times each in random order for 2 s each (see example, *Figure 3A*).

We confirmed that the spontaneous rate and response to stationary stimuli was significantly higher in females compared to males for OFS DN2 but not for OFS DN1 (triangles, *Figure 2—figure supplement 2A, B*, two-way ANOVA, p = 0.18 and p < 0.001, OFS DN1 and OFS DN2, respectively). As above, we subtracted the average response to the stationary stimulus from the responses to moving optic flow. The resulting data show that both neuron types exhibit sexual dimorphism to certain types of optic flow (*Figure 3E, F* and *Table 1*). For example, the OFS DN1 response to thrust shows a significant interaction between sex and velocity, with female neurons responding stronger to positive thrust compared to males (*Figure 3E* and *Table 1*). The OFS DN2 response to sideslip was significantly different between males and females, and there was a significant interaction between sex and velocity in the responses to sideslip, lift, and thrust, with males responding stronger to sideslip, lift, and thrust (*Figure 3F* and *Table 1*).

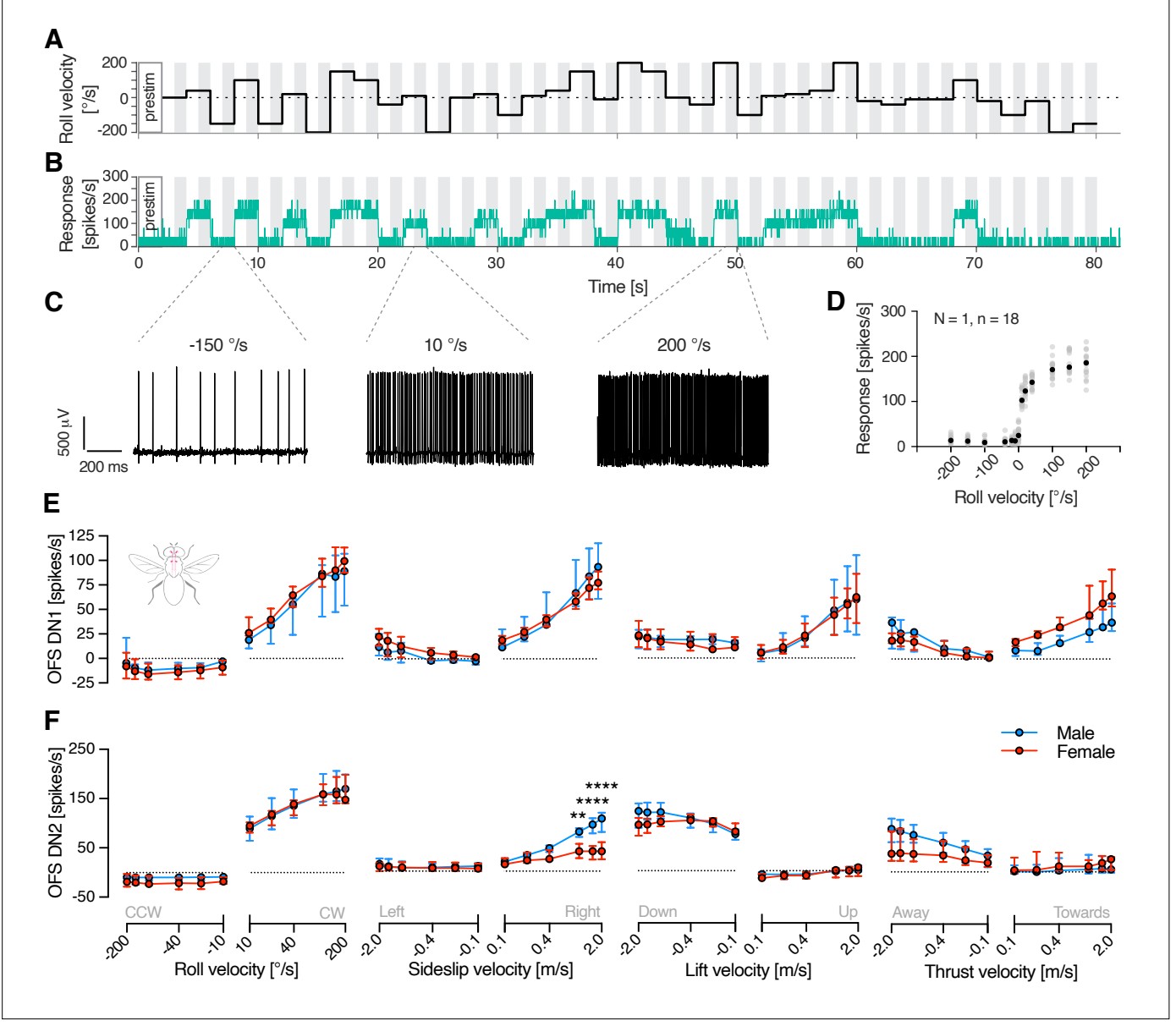

**Figure 3.** Velocity response functions in male and female OFS DNs. (**A**) Example stimulus profile over time, with roll velocity on the *y*-axis. (**B**) Representative spike histogram from a single trial from a male OFS DN2, smoothed using a 100 ms square-wave filter with 0.025 ms resolution, and time-aligned to the stimulus shown in panel A. Grey shading in panels A and B highlight the analysis windows used to calculate response. (**C**) Example extracellular raw data traces extracted from the analysis windows in panel B, illustrating neuronal responses to roll velocities of –150, 10, and 200°/s. (**D**) Average spike frequency (grey circles) was calculated for each repetition from each neuron (*N* = 1 neuron, *n* = 18 repetitions), and the median of these (black) was used for further analysis. (**E**) Velocity response functions of OFS DN1 in male (blue) and female (red) hoverflies in response to roll (*N* = 4 males, 5 females), sideslip (*N* = 5, 6), lift (*N* = 4, 4), and thrust (*N* = 3, 5). (**F**) Velocity response functions of OFS DN2 to roll (*N* = 10 males, 8 females), sideslip (*N* = 6, 7), lift (*N* = 6, 7), and thrust (*N* = 6, 7). Data in panels E and F are presented as median and interquartile range. Asterisks indicate statistically significant differences, two-way ANOVA with Šídák's multiple comparisons test (**p < 0.01 and ****p < 0.0001), see also *Table 1*.

## Morphological reconstruction suggests optic flow sensitive descending neurons could control the wings

By recording from the descending neurons intracellularly we could iontophoretically fill them with 3% neurobiotin following identification based on receptive field (as in *Figure 1*). We found that OFS DN1 and OFS DN2 of either sex receive their input in the part of the brain where LPTCs have their output (*Figure 4A, B, D–F*). Both neuron types project along the length of the thoracic ganglion, with several fine branches along the way (*Figure 4A–C, G–I*), like *Drosophila* DNHS1 and DNOVS2 (*Suver*

**Table 1.** Statistical summary of neural velocity response functions.
Results from two-way ANOVA analyses evaluating the effects of stimulus velocity and sex on neural responses (OFS DN1 and DN2). Asterisks denote levels of statistical significance: ***$p < 0.001$, ****$p < 0.0001$.

| Stimulus | | Velocity | Sex | Interaction | | Velocity | Sex | Interaction |
|---|---|---|---|---|---|---|---|---|
| Roll | | **** | 0.35 | 0.14 | | **** | 0.31 | >0.99 |
| Sideslip | OFS DN1 | **** | 0.90 | 0.12 | OFS DN2 | **** | 0.01 | **** |
| Lift | | 0.002 | 0.86 | >0.99 | | **** | 0.18 | *** |
| Thrust | | **** | 0.11 | **** | | **** | 0.10 | 0.02 |

et al., 2016). In *Drosophila*, DNOVS2 projects more medially than DNHS1, which also has a more distal projection, close to the haltere nerve (*Suver et al., 2016*). However, we saw no such differences between the projections of OFS DN1 and DN2 (*Figure 4A–C, G–I*). In addition, both hoverfly neuron types have prominent outputs where the prothoracic and pterothoracic nerves likely get their inputs (T1 LN and PtN, *Figure 4A–C, G–I*), which is more similar to the outputs of *Drosophila* DNOVS1 (*Suver et al., 2016*). These outputs suggest that OFS DN1 and DN2 could contribute to controlling the neck, wings and/or the forelegs.

While there was no indication of extensive sexual dimorphism in the structural morphology of these neurons, OFS DN2 appears to be slightly wider along the length of the cervical connective in females (*Figure 1—figure supplement 1A, B*). In part, this may be due to female hoverflies being bigger (*Daňková et al., 2023*; *Nicholas et al., 2018b*) and therefore having a wider cervical connective (*Figure 1—figure supplement 1C*).

## WBA changes are velocity dependent but not sexually dimorphic

Given that the morphological data suggest that both neuron types could control the wings (*Figure 4*), we conducted behavioural experiments in an open-loop tethered flight arena using the same continuous velocity stimulus as in electrophysiology (see *Figure 3A*). We tracked the WBA for each wing (*Figure 5A, B*, *Video 2*) using DeepLabCut models (*Mathis et al., 2018*; *Ogawa et al., 2025*) while the tethered animal was viewing different starfield velocities (*Figure 5C–F*). Changes in the sum of the left and right WBA (WBAS, *Figure 5B*) reflect variations in flight force generation, potentially contributing to either lift or thrust, while the difference between left and right WBA (WBAD) indicates yaw turning behaviour (*Götz, 1968*; *Maimon et al., 2010*; *Tammero et al., 2004*) or adjustments to body orientation (*Ristroph et al., 2010*).

The male WBAS is higher when viewing stationary stimuli compared with pre-stimulation ('pre stim' in *Figure 5C, D*; blue, *Figure 2—figure supplement 2C*, two-way ANOVA followed by uncorrected Fisher's LSD test, p < 0.01). However, there was no sexual dimorphism in the WBAS during pre-stimulation, nor when viewing stationary stimuli (*Figure 2—figure supplement 2C*, two-way ANOVA, p = 0.35). When comparing responses to each type of optic flow, after subtracting the response to stationary stimuli, both the WBAD and WBAS depended strongly on velocity; however, there was no sexual dimorphism (*Figure 5G, H* and *Table 2*).

The lack of sexual dimorphism observed in WBA (*Figure 5G, H* and *Table 2*) contrasts with the neural responses (*Figure 3E, F* and *Table 1*). Additionally, it is difficult to align the WBA responses to different types of optic flow with the neuronal input. For example, in response to higher clockwise roll velocities of (100–200°/s), hoverflies exhibit a decreased WBAD (positive roll, *Figure 5G*) and a slightly increased WBAS (positive roll, *Figure 5H*) indicating either a leftward turn or an anticlockwise body correction. Similarly, both OFS DN1 and DN2 give strong responses to clockwise roll (positive roll, *Figure 3E, F*). However, sideslip moving leftwards also elicits a decreased WBAD (negative sideslip, *Figure 5G*) and an increased WBAS (negative sideslip, *Figure 5H*) but neither neuron type responds strongly to this (negative sideslip, *Figure 3E, F*).

We found that neither lift nor thrust evoke large turning responses (*Figure 5G*); however, large and significant increases in the WBAS were observed in hoverflies experiencing the sensation of falling (positive lift, *Figure 5H* and *Table 2*) or being pushed backwards (negative thrust, *Figure 5H* and *Table 2*). Interestingly, each neuron type responds to lift and thrust in a different manner, with

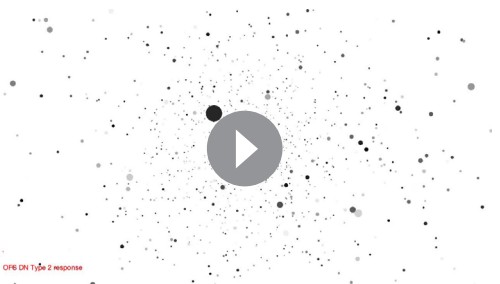

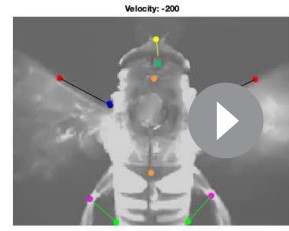

**Video 1.** Neural response to roll optic flow. The video shows the raw extracellular response of an OFS DN2 in red when the hoverfly was viewing an example roll stimulus, as in *Figure 3*.
https://elifesciences.org/articles/109795/figures#video1

**Video 2.** Female behavioural response to sideslip optic flow. The video on the left shows a female hoverfly filmed from above, with coloured dots indicating the body parts tracked by DeepLabCut to extract wing beat amplitude (WBA), head angle, and fore- and hind leg kinematics. The stimulus viewed by the hoverfly is shown on the right, with stimulus velocity indicated above the hoverfly video. The movie relates to *Figure 5*.
https://elifesciences.org/articles/109795/figures#video2

OFS DN1 being excited by upwards lift and approaching thrust (*Figure 3E*), whereas OFS DN2 is excited by downwards lift and receding thrust (*Figure 3F*).

In electrophysiology we used a horizontally oriented visual monitor (*Figure 5—figure supplement 1A*), whereas in behaviour the monitor was rotated 90° (*Figure 5—figure supplement 1B* and *Ogawa et al., 2025*). To ensure that this did not substantially affect the shape of the velocity-response functions, we reduced the stimuli to the central square in both behaviour and electrophysiology. This reduction caused a slight but significant change in neuronal (central square; *Figure 5—figure supplement 1C*, two-way ANOVA, p = 0.04) and WBAS responses (central square; *Figure 5—figure supplement 1D*, two-way ANOVA, p = 0.04). However, it did not alter the shape of the velocity response function nor produce a significant interaction between velocity and screen size (two-way ANOVA, p = 0.36 and 0.49, OFS DN2 and WBAS, respectively).

## Response onset depends on stimulus type

It has been previously shown that while neuronal responses are fast (*Leibbrandt et al., 2021*) behavioural responses occur on a much slower time scale and differ depending on stimulus type (*Schnell et al., 2014*; *Theobald et al., 2010*). We here found that there is no significant effect of sex on either neuronal or WBAS response onset (*Figure 6*, two-way ANOVA, p = 0.21, 0.79, and 0.31, OFS DN1, OFS DN2, and WBAS, respectively). However, the onset for roll compared to lift is significantly longer for OFS DN2 and WBAS but not for OFS DN1 (*Figure 6*, two-way ANOVA, p < 0.0001 for both OFS DN2 and WBAS compared to p = 0.08 for OFS DN1). Note that we recorded the WBA from above (*Figure 5A*), and as such did not quantify wing rotations and adjustments outside of the horizontal plane. The WBA response onset to roll optic flow (*Figure 6C*) should thus be viewed as highly conservative.

## Sideslip and lift optic flow generate strong head, fore-, and hind leg movements

The lack of sexual dimorphism in WBA suggests that behavioural responses to optic flow are the same between the sexes. To investigate this further, we next extracted the head angle (*Figure 7A, B*), and the extension of the fore- (*Figure 7A, C*) and hind legs (*Figure 7A, D*). We found that the head turned significantly when viewing sideslip (*Video 2*, *Video 3*, *Figure 7E*, and *Table 2*, two-way ANOVA, p < 0.001 for velocity), but there was no response to the other three types of optic flow. We found no sexual dimorphism (*Figure 7E* and *Table 2*).

We found that while the forelegs could not be seen in the dorsal view for most of the flight (see e.g. *Figure 5A*), they extended anteriorly in response to high velocity sideslip, lift, and thrust, but not to roll (*Figure 7F*, *Table 2*, and *Videos 2–9*). The foreleg extension to sideslip was sexually dimorphic, with significant interaction between sex and velocity for sideslip and lift motions (*Figure 7F* and *Table 2*, two-way ANOVA, p = 0.04, 0.01, and 0.03, respectively). Post hoc pairwise comparisons

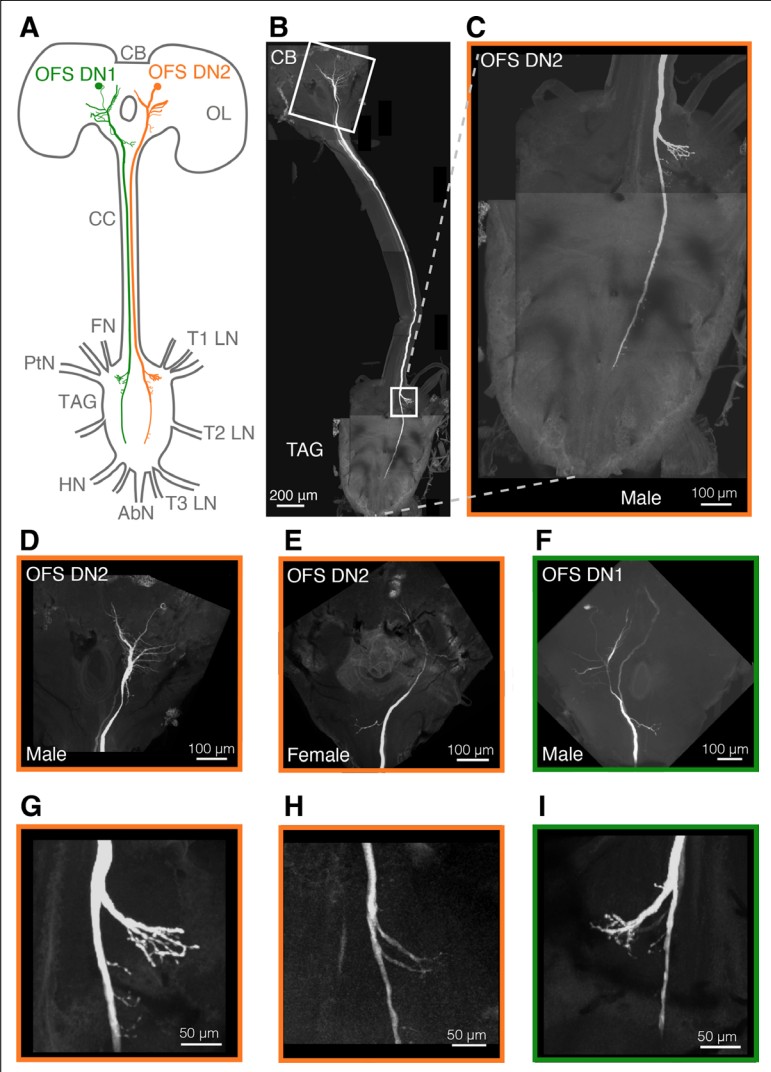

**Figure 4.** Morphological reconstruction of OFS DNs. (**A**) Schematic diagram of the hoverfly central brain and thoracic ganglia showing the projections of OFS DN1 (green) and OFS DN2 (orange). Key anatomical landmarks are labelled: CB, central brain; OL, optic lobe; CC, cervical connective; T1 LN, prothoracic leg nerve; T2 LN, mesothoracic leg nerve; T3 LN, metathoracic leg nerve; TAG, thoracic-abdominal ganglion; FN, frontal nerve; PtN, pterothoracic nerve; HN, haltere nerve; AbN, abdominal nerve. (**B**) Confocal image of a reconstructed male OFS DN2. White boxes indicate regions magnified in panels D–I. (**C**) A magnification of the thoracic ganglion in panel B. (**D**) Input dendrites of the same OFS DN2 around the sub-oesophageal ganglion. (**E**) Input dendrites of a female OFS DN2. (**F**) Input dendrites of a male OFS DN1. (**G**) Output projections of the same OFS DN2 neuron as in panels B–D. (**F**) Output projections within the thoracic ganglia of the neuron in panel E. (**H**) Output projections of the neuron in panel F.

The online version of this article includes the following figure supplement(s) for figure 4:

**Figure supplement 1.** Morphological quantification of OFS DN2.

---

revealed females extended their forelegs further than males to upwards lift at 1.5 m/s (*Figure 7F*, Šídák's multiple comparisons test, p = 0.01).

The hind legs were generally extended during flight (*Figures 5A, 7A*) with males extending their legs further than females (*Figure 2—figure supplement 2D*, two-way ANOVA, p = 0.032). After subtracting the hind leg extension when viewing the stationary starfield (filled symbols, *Figure 2—figure supplement 2D*), we found similar patterns to foreleg extension (*Figure 7F*), with a significant interaction between sex and velocity only in response to sideslip (*Figure 7G* and *Table 2*, two-way ANOVA, p = 0.02). We quantified the difference between the right and the left hind leg and found that

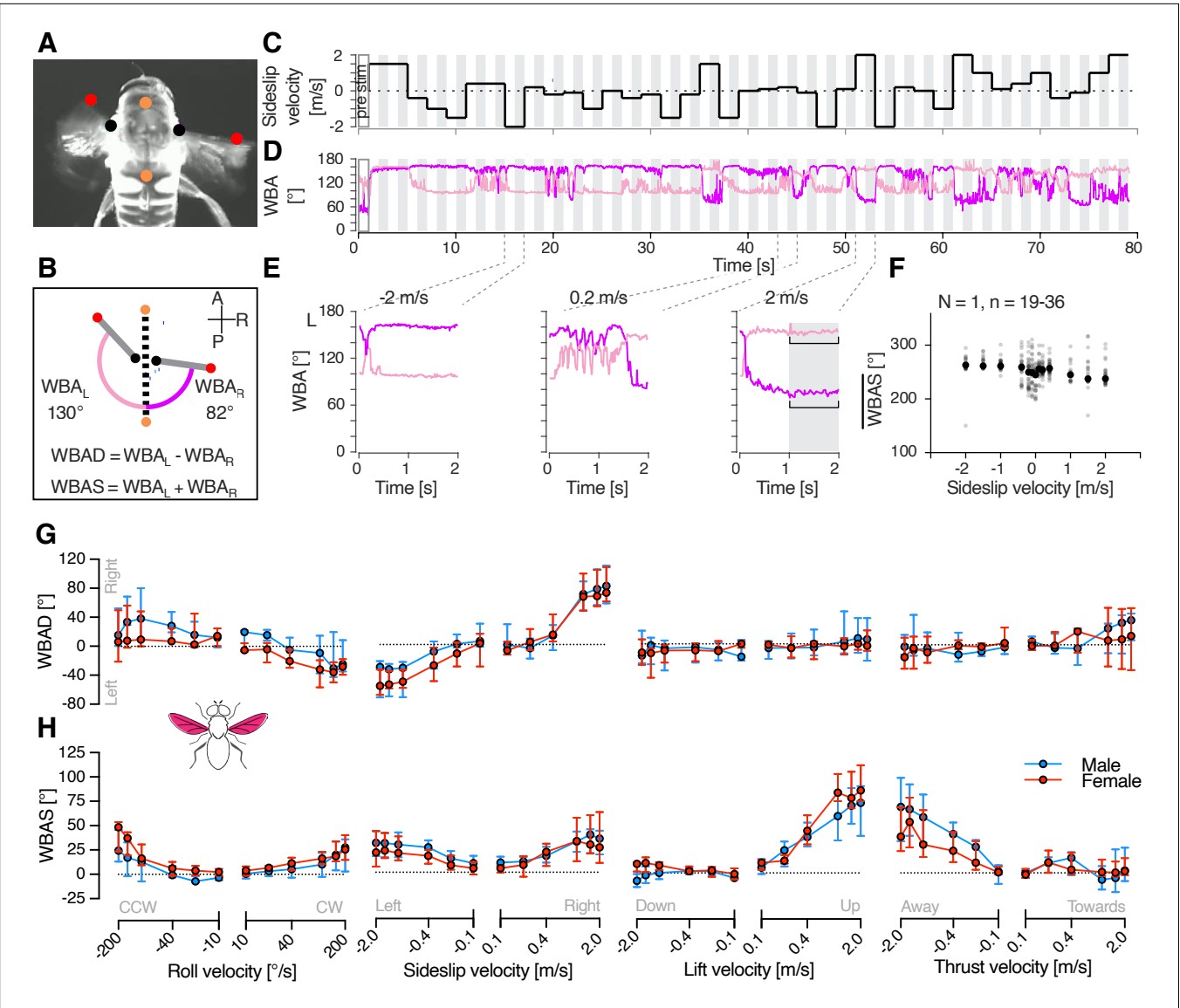

**Figure 5.** Wing beat amplitude (WBA) velocity response functions in male and female hoverflies. (**A**) A single frame from a video labelled with trained DeepLabCut models (*Mathis et al., 2018*; *Ogawa et al., 2025*; *Kane et al., 2020*). Coloured dots correspond to those in panel B, used to extract the WBA. (**B**) Pictogram illustrating a higher wing beat amplitude on the left wing (WBA_L) compared to the right wing (WBA_R), suggesting a turn to the right. Equations used to calculate wing beat amplitude difference (WBAD) and wing beat amplitude sum (WBAS). (**C**) Example stimulus with sideslip velocity on the y-axis. (**D**) Representative WBA for the left (pale colour) and right (salient colour) wings from a single trial, time-aligned with the stimulus shown in panel C. Grey shading in panels C and D indicate the analysis windows used to calculate WBAD and WBAS. (**E**) Magnified view of example WBA, showing behavioural responses to sideslip velocities of –2, 0.2, and 2 m/s. (**F**) Average WBAS (grey circles) calculated across repetitions (N = 1 animal, n = 19–36 repetitions). Black circles represent the median WBAS per stimulus velocity, used for further analysis. (**G**) WBAD in male (blue) and female (red) hoverflies in response to different optic flow velocities: roll (N = 5 males, 7 females), sideslip (N = 6, 6), lift (N = 7, 5), and thrust (N = 6, 5). (**H**) WBAS to the same stimuli in the same animals. Data in panels G and H are presented as median and interquartile range.

The online version of this article includes the following figure supplement(s) for figure 5:

**Figure supplement 1.** Impact of stimulus size on neural and behavioural velocity response functions.

there was a significant interaction between sex and velocity in response to roll and thrust (*Figure 7H* and *Table 2*, two-way ANOVA, p = 0.02 for roll and p < 0.0001 for thrust).

 Looking at the different body parts together we noted that in response to upwards lift (generating a falling sensation) the WBAS increases substantially (*Figure 5H*), and the fore- (*Figure 7F*) and hind legs (*Figure 7G*) all extend, potentially to brace for impact (*Videos 4 and 5*). We also noted that in response to sideslip, the WBAS increases (*Figure 5H*), the WBAD indicates a turn in the direction of

**Table 2.** Statistical summary of behavioural velocity response functions.

Results from two-way ANOVA analyses evaluating the effects of stimulus velocity and sex on behavioural responses (WBAS, WBAD, head angle, foreleg extension, hind leg extension, and hind leg difference). Asterisks denote levels of statistical significance: **p < 0.01, ***p < 0.001, ****p < 0.0001, after performing a Bonferroni correction (*Abdi, 2007*) for six comparisons of data from the same videos.

| Stimulus | | Velocity | Sex | Interaction | | Velocity | Sex | Interaction |
|---|---|---|---|---|---|---|---|---|
| Roll | | **** | 0.66 | >0.99 | | **** | 0.78 | >0.99 |
| Sideslip | WBAD | **** | >0.99 | >0.99 | WBAS | **** | >0.99 | >0.99 |
| Lift | | 0.18 | >0.99 | >0.99 | | **** | 0.54 | 0.12 |
| Thrust | | >0.99 | >0.99 | >0.99 | | ** | >0.99 | >0.99 |
| Roll | | 0.94 | >0.99 | >0.99 | | 0.89 | >0.99 | 0.26 |
| Sideslip | Head | ** | >0.99 | >0.99 | Foreleg | ** | **0.04** | **0.01** |
| Lift | | >0.99 | >0.99 | >0.99 | | **** | 0.38 | **0.03** |
| Thrust | | >0.99 | >0.99 | 0.08 | | ** | >0.99 | >0.99 |
| Roll | | 0.21 | >0.99 | 0.17 | | ** | >0.99 | **0.02** |
| Sideslip | Hind leg | **0.04** | >0.99 | **0.02** | ΔHind leg | **** | >0.99 | >0.99 |
| Lift | | 0.39 | >0.99 | >0.99 | | 0.41 | >0.99 | >0.99 |
| Thrust | | 0.07 | >0.99 | >0.99 | | ** | 0.06 | **** |

the optic flow (*Figure 5G*), the head rotates (*Figure 7E*), and both the forelegs and hind legs extend (*Figure 7F, G*). The hind legs appear to be ruddering, with the right leg extending more when the animal is turning right (*Figures 5G and 7H*, *Videos 2 and 3*). There was no significant difference in the response onset of these body parts, nor was there any sexual dimorphism (*Figure 7—figure supplement 1*).

## Discussion

In the hoverfly, the optics, photoreceptors, and HS cells exhibit clear sexual dimorphism (*Straw et al., 2006*; *van Hateren et al., 1989*; *Nordström et al., 2008*; *Barnett et al., 2010*). In line with this, our findings reveal that optic flow sensitive descending neurons also show sexually dimorphic velocity response functions (*Figure 3*). Yet, despite the multi-level anatomical and neural sexual dimorphism, the WBA (*Figure 5*) and head movements (*Figure 7*) remain mostly monomorphic, while leg movements exhibit clear sex-specific differences. Moreover, differences between neural and behavioural velocity response functions (*Figures 3, 5, and 7*) and response latency (*Figure 6*) highlight the

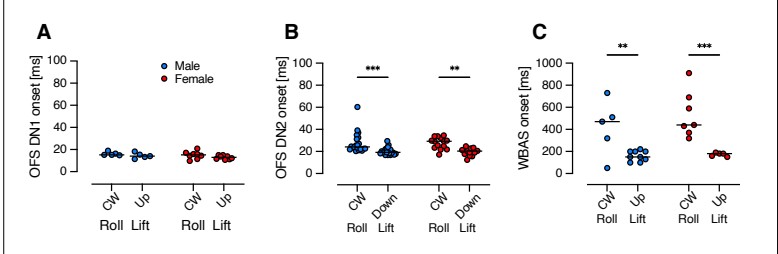

**Figure 6.** Response onset to roll and lift stimuli in male and female hoverflies. (**A**) Time to response onset in OFS DN1 to roll (+50°/s) or lift (+0.5 m/s) stimuli in males (blue, N = 5) and females (red, N = 8). (**B**) Response onset in OFS DN2 to roll (+50°/s) or lift (–0.5 m/s) measured in males (N = 20) and females (N = 14). (**C**) Onset of WBAS responses to roll (–200°/s) and lift (+2 m/s) in males (N = 5 for roll, 9 for lift) and females (N = 7 for roll, 5 for lift). Data are presented from individual neurons (**A, B**) or animals (**C**) with lines indicating median. Asterisks indicate statistically significant differences, two-way ANOVA with uncorrected Fisher's LSD test (**p < 0.01 and ***p < 0.001).

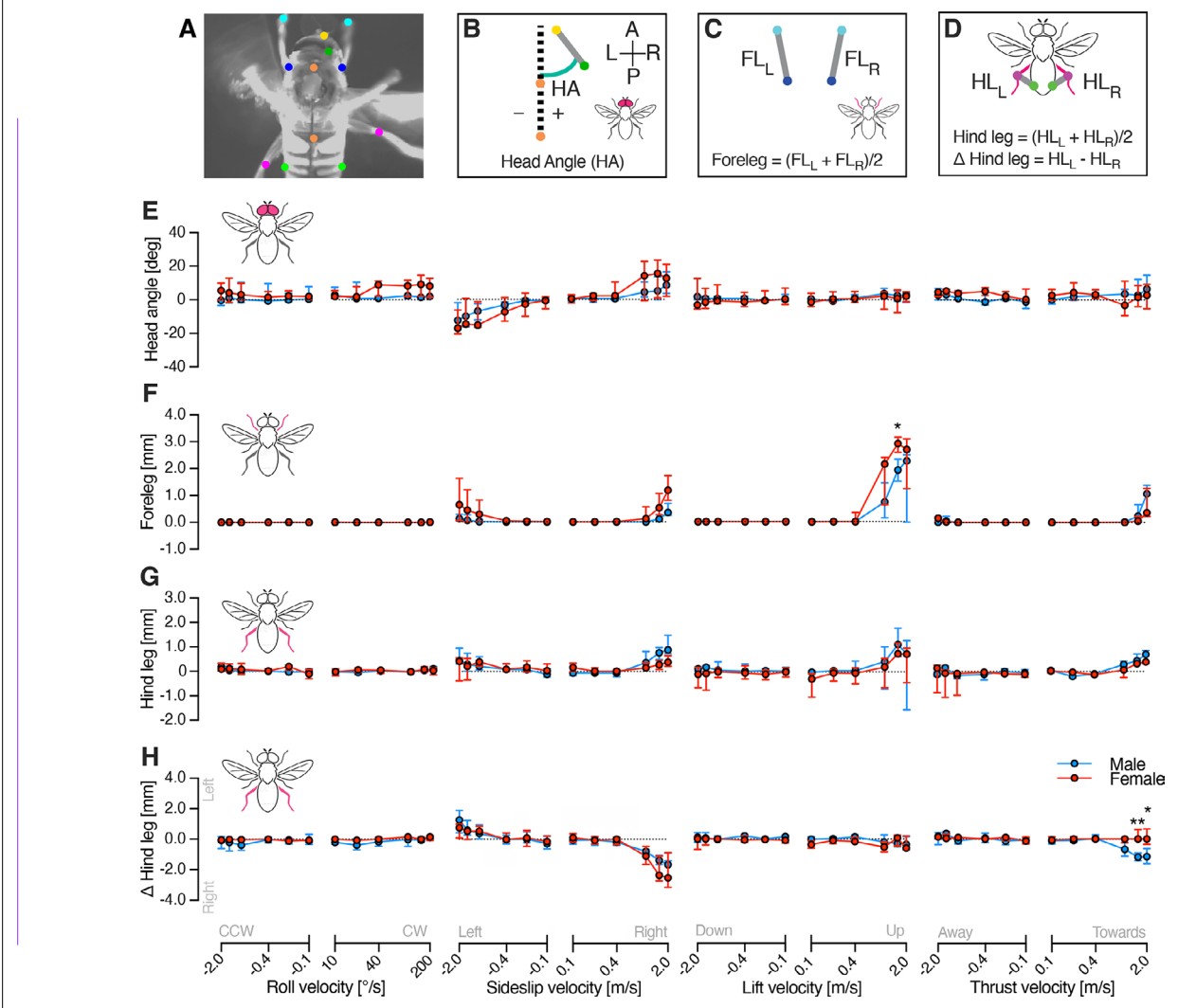

**Figure 7.** Velocity response functions of the head, fore-, and hind legs in male and female hoverflies. (**A**) A single frame highlighting the points used to track the head (yellow and dark green), thorax (orange), forelegs (pale and dark blue), and hind legs (magenta and pale green). (**B**) The head angle (HA) was defined as the angle between the medio-posterior head and the longitudinal thorax axis. (**C**) The foreleg extension was defined as the distance between the proximal and distal point, after calculating the average for the left and the right leg. (**D**) The hind leg extension was defined as the distance between the knee and a lateral point on the mid-abdomen, after calculating the average for the left and the right leg. We also calculated the difference between the right and the left hind leg. (**E**) The head angle in male (blue) and female (red) hoverflies in response to different optic flow velocities: roll ($N$ = 5 males, 7 females), sideslip ($N$ = 6, 6), lift ($N$ = 7, 5), and thrust ($N$ = 6, 5). (**F**) The foreleg extension to the same stimuli in the same animals. (**G**) The hind leg extension in the same animals. (**H**) The difference between the left and right hind leg extension in the same animals. The data in panels E–H are presented as median and interquartile range. Asterisks indicate statistically significant differences, two-way ANOVA with Šídák's multiple comparisons test (*$p < 0.05$, **$p < 0.01$), see also **Table 2**.

The online version of this article includes the following figure supplement(s) for figure 7:

**Figure supplement 1.** Response onset to sideslip.

likely involvement of downstream processing mechanisms that could reconcile sex-specific sensory encoding with conserved flight control.

Some hoverfly flight behaviours are strongly sexually dimorphic, such as those related to courtship and mating. For example, male hoverflies defend territories from a hovering stance from which they pursue and capture female conspecifics at high speeds (*Collett and Land, 1978*). In contrast, other behaviours, including cruising speeds in field (*Thyselius et al., 2018*) and indoor settings (*Thyselius et al., 2023*) do not differ between sexes and are much slower (median around 0.3 m/s; *Thyselius et al., 2018*, *Thyselius et al., 2023*) than the 10 m/s that can be reached during outdoor pursuit

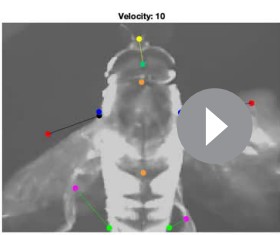

**Video 3.** Male behavioural response to sideslip optic flow. The video shows a male hoverfly filmed from above, viewing sideslip at different velocities.
https://elifesciences.org/articles/109795/figures#video3

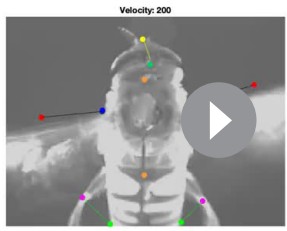

**Video 4.** Female behavioural response to lift. A female hoverfly filmed from above, viewing lift optic flow at different velocities.
https://elifesciences.org/articles/109795/figures#video4

(*Collett and Land, 1978*). In comparison, our results show limited sex-related differences in WBA in response to optic flow speeds up to 2 m/s (*Figure 5*), yet we detected pronounced sexual dimorphism in neural responses at velocities as low as 0.5 m/s (*Figure 3*, *Table 1*). This suggests that sensory processing differences emerge at lower velocities than the velocities where motor output diverges. Moreover, the WBAS response latencies were much longer than the corresponding neural activity (*Figure 6*), implying that temporal integration or additional circuit-level modulation may delay and refine motor execution. Together, these findings suggest a complex transformation between sex-specific sensory encoding and conserved motor behaviour, potentially mediated by downstream integration or additional alternate neural pathways that bypass the descending neurons studied here.

In *Drosophila* there are around 1000 descending neurons, of which at least 35 types receive their input in the posterior surface of the brain, which includes the inferior posterior slope, where the LPTC outputs are found (*Wei et al., 2020*; *Strausfeld and Bassemir, 1985a*; *Namiki et al., 2018*). However, only a subset of these constitute the 29 descending neuron types that project to the wing motor neuropil (*Namiki et al., 2018*). It is thus likely that the WBA that we recorded here (*Figure 5*) reflects contributions from descending neurons beyond the two that we have characterised (*Figures 1 and 3*) in this study. For example, many looming sensitive neurons also respond to widefield optic flow (*Nicholas et al., 2020a*; *Nicholas et al., 2023*) and could therefore play a role in the WBA changes that we detected (*Figure 5*). While the morphology of both neurons (*Figure 4*) suggested involvement in wing motor control, there is currently no direct evidence implicating them in the control of wingbeat amplitude. Interestingly, DNOVS2 in *Drosophila*, a physiologically similar neuron to OFS DN2 (*Nicholas et al., 2020a*), has been indirectly linked to rapid turning behaviour (*Suver et al., 2016*). Furthermore, silencing HS and VS cells, the presumed presynaptic LPTCs to these descending neurons, results in reductions in WBA only at higher stimulus speeds (*Kim et al., 2017*). These findings suggest that HS and VS cells, along with their downstream targets, may be specialized for driving fast optomotor responses under high-speed visual motion (>180°/s), rather than broadly regulating WBA across all velocities, including those examined in this study.

Beyond their potential involvement in wingbeat amplitude modulation, OFS DNs may play a role in coordinating head and body positioning during flight, especially in contexts requiring precise visual

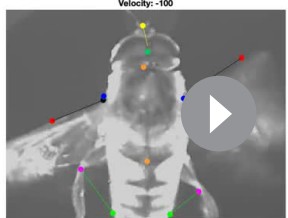

**Video 5.** Male behavioural response to lift. A male hoverfly filmed from above, viewing lift optic flow at different velocities.
https://elifesciences.org/articles/109795/figures#video5

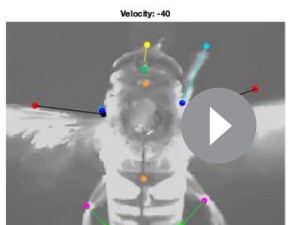

**Video 6.** Female behavioural response to thrust. A female hoverfly filmed from above, viewing thrust optic flow at different velocities.
https://elifesciences.org/articles/109795/figures#video6

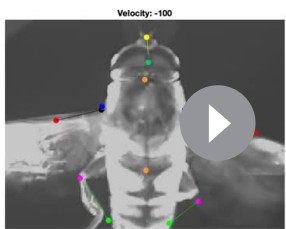

**Video 7.** Male behavioural response to thrust. A male hoverfly filmed from above, viewing thrust optic flow at different velocities.

https://elifesciences.org/articles/109795/figures#video7

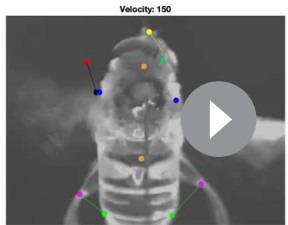

**Video 8.** Female behavioural response to roll. A female hoverfly filmed from above, viewing roll optic flow at different velocities.

https://elifesciences.org/articles/109795/figures#video8

alignment and rapid manoeuvring. Indeed, the *Drosophila* physiological homologs of OFS DN1 and OFS DN2 (**Nicholas et al., 2020a**), DNHS1 and DNOVS2, project to the neck motor neuropil (**Namiki et al., 2018**), and have been implicated in controlling head movements, abdominal ruddering, and engagement with the haltere motor system for flight stabilization (**Suver et al., 2016**). While we did not quantify abdominal or haltere movements, we found that the head turned when viewing sideslip optic flow (**Figure 7E**), and that the velocity dependence was similar to neural (**Figure 3E**) and WBAD (**Figure 5G**) responses. While we did not detect any head movements to roll, lift or thrust, this could be a technical limitation of recording from above (**Figure 5A**, **Videos 4–9**). Indeed, many of the 29 descending neuron types that project to the wing motor neuropil also project to the neck motor neuropil (**Namiki et al., 2018**), suggesting that synchronized responses are not unexpected. Neither are the similar responses of the fore- and hind legs surprising (**Figure 7F, G**), considering that in *Drosophila* many of the ~30 descending neuron types that project to the forelegs also project to the mid- and hind legs (**Namiki et al., 2018**).

Our WBA recordings would also be affected by filming from above and thus lacking other nuanced changes in wing angles. Thus, the sexual dimorphism in neural responses (**Figure 3**) could reflect an evolutionary tuning of visuomotor pathways in males, optimized for fast, directional adjustments rather than gross changes in WBA in the horizontal plane. For instance, while changes in WBAS are similar when generating either lift or thrust (**Figure 5H**), body pitch dynamically adjusts their ratio, driving behavioural variation (**Zanker, 1990**). Thus, OFS DNs may serve as integral components of a broader flight control architecture, interfacing optic flow detection with dynamic body and head positioning systems to support complex, sex-specific behavioural outcomes, particularly in males engaging in high-speed pursuits.

Indeed, flight speed in insects results from a complex integration of multiple kinematic parameters, not solely from changes in wingbeat amplitude. Many species refine their aerodynamic output by modulating wingbeat frequency, angle of attack, wing tip trajectory, deviations from the mean stroke plane and through precise adjustments to the timing and duration of the up- and downstrokes (**Sane, 2003**). These control strategies support agile manoeuvring, particularly during visually guided behaviours such as the high-speed pursuits undertaken by male hoverflies. Furthermore, the smaller body size of male hoverflies compared to females (**Daňková et al., 2023**; **Nicholas et al., 2018b**) may confer biomechanical advantages, including reduced mass, facilitating faster acceleration, heightened responsiveness, and lower metabolic costs for executing flight manoeuvres (**Dudley, 2002**; **Niven and Scharlemann, 2005**), which do not appear in a tethered flight set-up. Such enhanced agility is likely crucial for rapid direction changes required during courtship and may enable male hoverflies to outperform female flies without relying on increased wingbeat amplitude.

Taken together, our findings reveal significant differences between sexually dimorphic sensory encoding and conserved motor output in hoverfly

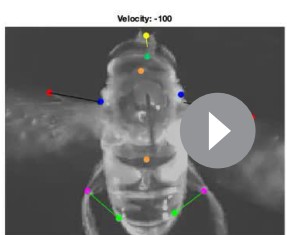

**Video 9.** Male behavioural response to roll. A male hoverfly filmed from above, viewing roll optic flow at different velocities.

https://elifesciences.org/articles/109795/figures#video9

flight at cruising speeds. Although OFS DNs and their upstream visual circuits display clear sex-specific tuning, wingbeat amplitude and head angle changes in response to optic flow stimuli remain relatively similar between the sexes, suggesting additional mechanisms downstream of sensory input. This likely reflects a complex interplay of biomechanical properties, multisensory integration and circuit-level modulation, each shaping and refining behavioural outcomes to meet distinct demands, preserving the consistency of low-speed manoeuvres while enabling sex-specific tuning during high-speed pursuits.

## Methods

### Key resources table

| Reagent type (species) or resource | Designation | Source or reference | Identifiers | Additional information |
|---|---|---|---|---|
| Biological sample (hoverfly) | *Eristalis tenax* | From wild-caught females | RRID:NCBITaxon_2061809 | For details, see *Nicholas et al., 2018b* |
| Software, algorithm | DeepLabCut (DLC) | See *Mathis et al., 2018* | | Version 2.3.3 |
| Software, algorithm | MATLAB | Mathworks | | |
| Software, algorithm | FlyFly | https://github.com/HoverflyLab/FlyFly, *HoverflyLab, 2025* | | Visual stimulus generation |
| Software, algorithm | Psychophysics toolbox for MATLAB | See *Brainard, 1997*; *Pelli, 1997* | | |
| Software, algorithm | Circstat toolbox for MATLAB | See *Berens, 2009* | | |
| Software, algorithm | Prism | GraphPad Software | | Version 10.4.0 |

### Animals

For all experiments, male and female *E. tenax* were reared and housed as described previously (*Nicholas et al., 2018b*). Briefly, eggs were collected from females captured under permit in Wittunga Botanic Garden, Adelaide, South Australia. Upon hatching, larvae were reared in a rabbit dung slurry until third instar larvae emerged to pupate. Eclosion occurred 1–2 weeks post-pupation. Adult hoverflies were used for behavioural experiments at 17–87 days post-eclosion, for intracellular electrophysiology and subsequent morphological reconstruction at 38–54 days, and for extracellular electrophysiology at 8–204 days.

### Electrophysiology

Before recording the animal was immobilised, mounted dorsal side down (*Figure 5—figure supplement 1A*) and secured using a mixture of beeswax and resin. A small region of cuticle was removed at the anterior end of the thorax to expose the cervical connective. If required, any excessive gut or tracheal tissue obstructing the recording site was removed and a small volume of PBS was added to prevent drying within the ventral cavity. A fine wire hook was positioned under the cervical connective for mechanical support, and a silver wire was inserted into the cavity to serve as a reference electrode and grounding wire (*Nicholas et al., 2020a*).

For extracellular recordings, a sharp tungsten microelectrode (2 MΩ, polyimide-insulated; Microprobes) was inserted into the cervical connective (*Nicholas et al., 2020a*). Signals were amplified 1000 times and band-pass filtered between 10 and 3000 Hz using a DAM50 differential amplifier (World Precision Instruments), followed by noise reduction with a HumBug (Quest Scientific). Data were digitized via a PowerLab 4/30 interface (ADInstruments) and acquired at 40 kHz. Spike sorting was performed in LabChart 7 Pro (ADInstruments) based on the amplitude and width of individual action potentials.

For intracellular recordings, aluminosilicate electrodes were pulled using a Sutter P-1000 micropipette puller, achieving a resistance of approximately 40–70 MΩ. Electrode tips were filled with 3% neurobiotin (Vector Laboratories), then backfilled with 1 M KCl using a syringe, leaving a small air bubble between the two solutions. Electrodes were inserted into the cervical connective for recording, and the resulting signal was amplified using an Axoclamp-2B amplifier (Axon Instruments), followed by 50 Hz noise reduction with a HumBug (Quest Scientific). Data acquisition and digitization

were performed at 10 kHz using an NI USB-6210 16-bit data acquisition card (National Instruments) and the MATLAB Data Acquisition Toolbox (Mathworks), using in house software (https://github.com/HoverflyLab/SampSamp, copy archived at *HoverflyLab, 2026*).

## Morphological reconstructions

Following intracellular recordings, neurons were stained iontophoretically with neurobiotin using currents in the 1 nA range for 3–12 min. The nervous system was then carefully dissected and fixed in 4% paraformaldehyde overnight. Tissue was incubated with a Cy3-streptavidin conjugate (1:100; Jackson ImmunoResearch) for 2 hr, then dehydrated through an ethanol series (50–100%) for 15–20 min per step. After washing in PBT, the tissue was cleared in RapiClear (SUNJIN Lab) and mounted with spacers. Imaging was performed using a Zeiss LSM 880 Fast Airyscan confocal microscope at the institutional microscopy facility. Neuron morphology and cervical connective width were quantified using ImageJ (*Schneider et al., 2012*).

## Tethered flight

Prior to flight recordings, hoverflies were tethered at a 32° angle (*Figure 5—figure supplement 1B*) using a beeswax–resin mixture to a small pin inserted into a hypodermic needle (BD Microlance, 23G × 1¼"). Flight was initiated by manually providing airflow for 1–10 min until consistent flight behaviour was observed. Once positioned facing the centre of the visual monitor (*Figure 5—figure supplement 1B*), hoverflies were filmed from above at 100 Hz using a Sony PlayStation 3 Move Eye Camera (SLEH-00448, Sony) with the IR filter removed, and equipped with an infrared pass filter (R72 INFRARED, 49 mm, HOYA; for details see *Ogawa et al., 2025*). Illumination was provided by infrared LEDs inserted into USB lights (JANSJÖ LED USB lamp, IKEA) and a Musou Black (Shin Kokushoku Musou black, KOYO Orient Japan) surface was placed beneath the hoverfly to enhance contrast and minimize optical interference.

We used DeepLabCut (DLC) version 2.3.3 (*Mathis et al., 2018*) to train a model to track the thorax and the peak downstroke angle, referred to as the WBA, of the left and right wing (WBA$_L$ and WBA$_R$; *Figure 5A, B*), as described previously (*Ogawa et al., 2025*). In addition, we tracked the dorso-medial head (yellow, green, *Figure 7A, B*), the proximal and distal points of the forelegs (blue, *Figure 7A, C*), the hind leg knee and the lateral, mid-abdomen (magenta, green, *Figure 7A, D*). For this purpose, we first manually labelled the anatomical landmarks (*Figures 5A and 7A*), across 16 extracted video frames per individual from four hoverflies (2 males, 2 females). In addition to examples where the hoverfly was not flying, these frames included responses to yaw rotation and forward translation. The DLC model was trained for 300,000 iterations, yielding train and test errors of 1.2 and 1.16 pixels, respectively.

To identify potential tracking errors from DLC, we smoothed the WBA time series using MATLAB's *smooth* function with both *loess* and *rloess* methods. Because *rloess* is more resistant to outliers, we used it to detect abnormal data points. For each wing, if the absolute difference between the *loess*- and *rloess*-smoothed signals exceeded 5% for more than 1 s, the data were excluded due to suspected tracking artefacts. The smoothing was used only for error detection; all subsequent analyses were performed on the unsmoothed data. In addition, we excluded data if the WBA of either wing dropped below 40° for at least 0.5 s, as this indicated a cessation of flight. Finally, entire trials were excluded if the hoverfly was not flying for more than 50% of the trial duration. Head and leg kinematic data were excluded whenever the corresponding WBA data were excluded, ensuring consistent trial inclusion across behavioural metrics.

## Visual stimuli

Visual stimuli were generated using custom software (https://github.com/HoverflyLab/FlyFly, *HoverflyLab, 2025*) written in MATLAB, incorporating the Psychophysics Toolbox (*Brainard, 1997*; *Pelli, 1997*). All screens had a refresh rate of 165 Hz and a linearized contrast with a mean illuminance of 200 Lux. For intracellular recordings, the hoverfly was placed 13 cm away from a ViewSonic screen with a resolution of 2560 × 1440 pixels, corresponding to 143° × 107° of the visual field. For extracellular recordings, the hoverfly was placed 6.5 cm away from a 2560 × 1440 pixel Asus screen, yielding a projected visual field of 155° × 138° (*Figure 5—figure supplement 1A*). For behavioural recordings, the hoverfly was placed 10 cm from a vertically orientated Asus screen (1440 × 2560 pixel,

*Figure 5—figure supplement 1B*) producing a projected size of 118° × 142°. To evaluate the impact of different screen orientations in electrophysiology and behaviour, visual stimuli were presented either full-screen or using the central 1440 × 1440 pixel square (*Figure 5—figure supplement 1C, D*).

## Receptive field mapping

To map each neuron's receptive field, we presented local sinusoidal gratings (average 38° × 38°) drifting in 8 directions for 0.36 s each, across 48 overlapping locations, as described previously (*Nicholas et al., 2020a*). Stimuli were full contrast, with an average spatial frequency of 0.14 cycles/° and a temporal frequency of 5 Hz. At each location, we calculated the local maximum spike frequency (red, inset, *Figure 1B, E*). After subtracting the spontaneous rate, calculated for 0.8 s preceding stimulus onset (dotted line, inset, *Figure 1B, E*), we interpolated the resulting local maximum responses to a tenfold higher spatial resolution (colour coding, *Figure 1A, D*). We then quantified the 50% response maximum using MATLAB's *contour* function (black, *Figure 1B, E*). We created a polygon-shape of this 50% contour line using MATLAB's *polyshape* function to calculate the receptive field size, defined as its width and height (*Figure 1—figure supplement 2F*), and used the *centroid* function to identify the centre of this polyshape (red circle, *Figure 1B, E*). As recordings were performed with the animal ventral side up (*Figure 2—figure supplement 2A*, *Figure 5—figure supplement 1A*), receptive fields were rotated to display them with the dorsal visual field up.

Next, at each location, we fit a cosine function to the responses to the different directions of motion, where the LPD and LMS are the peak position and amplitude of the cosine fit, respectively (inset, *Figure 1C, F*). These were visualized as vectors where angle indicates LPD and length indicates LMS (arrows, *Figure 1A, D*). We defined the receptive field preferred direction as the median of LPDs at locations where LMS exceeded 50% of the maximum (black and red arrows; *Figure 1C, F*), using *circ_median* from the CircStat toolbox for MATLAB (*Berens, 2009*). We calculated the LPD variance using the *circ_var* functions from CircStat toolbox in MATLAB (*Berens, 2009*), of LPDs at locations where LMS exceeded 50% of the maximum (black and red arrows; *Figure 1C, F*).

We extracted unpublished receptive field data from 100 reference neurons (other data from five of these optic flow sensitive descending neurons has been published previously *Nicholas and Nordström, 2021*) to set exclusion criteria (red shading, *Figure 1—figure supplement 2A–C*) and to classify the OFS DNs used in the rest of this study. OFS DN1 on the LHS were defined by receptive field preferred directions between 120° and 200° (light green, *Figure 1G, I*). OFS DN1 on the RHS were defined by receptive field preferred directions between 340° and 60° (dark green, *Figure 1G, I*). OFS DN2 were classified by receptive field preferred directions between 220° and 320° (yellow and orange, *Figure 1G, I*), with the azimuthal location of the receptive field centre determining whether they were LHS or RHS (yellow and orange, *Figure 1H*).

A polar plot where the distance from origo indicates azimuthal position, and the location along the radius the preferred direction, confirms that the LHS and RHS of OFS1 and OFS2 cluster distinctly (*Figure 1—figure supplement 1A*). To test whether incorporating additional receptive field parameters would reveal further neuronal subtypes, we performed *k*-means clustering in MATLAB using *z*-score normalised data. The additional parameters included receptive field centre elevation, receptive field height and width (see *Figure 1—figure supplement 2F*), maximum LMS (see *Figure 1—figure supplement 2A*), the number of positions with LMS values exceeding 50% of the maximum (see *Figure 1—figure supplement 2B*), and LPD variance (see *Figure 1—figure supplement 2C*). However, the (*Caliński and Harabasz, 1974*) values indicate that adding additional receptive field parameters actually reduced clustering performance (*Figure 1—figure supplement 1B*). In contrast, clustering based solely on preferred direction and azimuthal location yielded four well-defined groups (*Figure 1—figure supplement 1C*) and produced the highest Callinski–Harabasz value (*Figure 1—figure supplement 1B*).

## Directional sensitivity

Full-screen sinusoidal gratings were presented at full contrast, using the same spatial (0.14 cycles/°) and temporal (5 Hz) frequencies as in receptive field mapping. For each stimulus direction, mean spike frequency was calculated over the 1-s stimulus duration, excluding the first 100 ms to avoid onset transients (*Nicholas and Nordström, 2020b*). We fit a cosine function to the responses to the eight different directions of motion, to extract the preferred direction and response amplitude

(*Figure 2—figure supplement 1B*), and then plotted these values for OFS DN1 (*Figure 2—figure supplement 1C*) and DN2 neurons (*Figure 2—figure supplement 1D*). To account for the ventral-side-up recording position (*Figure 2—figure supplement 1A*) directional responses were rotated 180° (*Figure 2—figure supplement 1F–H*). In addition, to display all neurons as RHS, and assuming mirror-symmetry, we flipped LHS neurons across the midline (*Figure 2—figure supplement 1I–L*).

## Responses to optic flow

We used a starfield stimulus to generate the type of perspective-corrected optic flow that would have been seen by the hoverfly if it was moving through a space of 2-cm diameter spheres (for details, see *Nicholas et al., 2020a*; *Leibbrandt et al., 2021*). These simulated translations at 0.5 m/s (sideslip, lift, and thrust) or rotations (pitch, yaw, and roll), at 50°/s. To quantify neural responses, the mean spike frequency was calculated over the 0.97 s stimulus duration, excluding the first 0.1 s to avoid onset-related transients (*Nicholas and Nordström, 2020b*). The spontaneous firing rate, averaged across 0.48 s immediately preceding stimulus onset (open circles, *Figure 2—figure supplement 2A, B*), was subtracted from the response.

## Velocity response functions

We used four types of optic flow: three translations (sideslip, lift, and thrust, *Videos 2–7*) and one rotation (roll, *Video 8*, *Video 8* and *Video 9*). Translations were presented at velocities of −2, −1.5, −1, –0.4, −0.2, −0.1, 0, 0.1, 0.2, 0.4, 1, 1.5, and 2 m/s, while rotations were presented at 200, −150, −100, –40, −20, −10, 0, 10, 20, 40, 100, 150, and 200°/s. The sign of the velocity indicates the direction of motion as seen by the fly when corrected for its position, with positive values corresponding to counterclockwise roll, leftward sideslip, downward lift, and thrust moving away. Conversely, negative values indicate clockwise roll, rightward sideslip, upward lift, and thrust moving toward the hoverfly. The upper limit was defined by the movement of the individual dots within the starfield stimulus between frames, that is it was limited by the refresh rate of the screen.

Each trial consisted of 39 stimuli (13 unique velocities × 3 repetitions), presented in a random order, with each stimulus lasting 2 s, immediately followed by the next velocity. Trials began with a 1-s blank screen, which served as the pre-stimulation baseline (*Figures 3A and 5C*). Each optic flow condition was repeated multiple times, resulting in at least nine repetitions for each velocity and neuron, and five repetitions for each velocity and animal in behaviour.

For neuronal recordings, velocity response function trials were interleaved with the optic flow stimuli described above. In behavioural experiments, a flight refresher sequence interspersed each velocity tuning trial. This consisted of sinusoidal gratings (200° wavelength, 5 Hz) drifting rightward, leftward, and rightward again for 4 s each.

Neural responses were quantified as the mean spike frequency during the final second of stimulation (grey shaded areas, *Figure 3A–C*). We then calculated the median across trials for each velocity (*Figure 3D*). Pre-stimulation activity (spontaneous rate) was measured over a 2-s window immediately preceding stimulus onset. Neural responses are displayed after subtracting the response when viewing a stationary stimulus (filled symbols, *Figure 2—figure supplement 2A, B*).

In behavioural experiments, we extracted the mean WBA of the left and right wings ($\overline{WBA_L}$, $\overline{WBA_R}$, *Figure 5B*) during the final second of stimulation (grey shaded areas, in *Figure 5C–E*). For each velocity, we calculated the WBA difference (WBAD, defined as $\overline{WBA_L} - \overline{WBA_R}$) and the WBA sum (WBAS, defined as $\overline{WBA_L} + \overline{WBA_R}$, *Figure 5B*). We then calculated the median across repetitions for each individual (*Figure 5F*). Pre-stimulation responses were averaged over a 0.5-s time window immediately preceding stimulus onset. WBA responses are displayed after subtracting the response when viewing a stationary stimulus (filled symbols, *Figure 2—figure supplement 2C*).

We defined the head angle as the angle between a straight line joining the 2 tracked points on the head (using *polyfit* in MATLAB, *Figure 7A, B*) and the longitudinal axis of the thorax (*Figure 7A, B*). As we filmed in one plane only, this measurement can be caused by a combination of head rotations but seems to be dominated by roll rotations (*Videos 2–9*). We measured the distance between the proximal and distal points of the forelegs (*Figure 7A, C*). Note that the forelegs were mostly hidden under the animal from our dorsal view and could only be seen when extended anteriorly (e.g. *Figure 5A*, see also *Videos 2–9*). Data are displayed as the mean extension of the left and right foreleg. We measured the distance between the hind leg knee and a lateral point on the mid-abdomen (*Figure 7A, D*). Hind

leg data are displayed as the mean or the difference of the left and right hind leg. All body parts data are displayed after subtracting the response when viewing a stationary stimulus (filled symbols show hindleg data, *Figure 2—figure supplement 2D*).

### Response onset

For neuronal recordings, we compared onset times for clockwise roll (+50°/s) and either upwards lift (+0.5 m/s) for OFS DN 1 or downwards lift (–0.5 m/s) for OFS DN2, stimuli which generated strong responses in these neurons (*Figure 3E, F*). Mean spike frequency was calculated over the 0.97 s stimulus duration, excluding the first 0.1 s to avoid onset transients (*Nicholas and Nordström, 2020b*). Onset was defined as the first time point after the first 0.1 s, where spike rate exceeded 80% of the mean.

For behaviour, we used WBA responses to roll (–200°/s) and lift (+2 m/s). Mean WBAS was defined as the average response during the final second of stimulation, and onset was defined as the first time point where WBAS exceeded 80% of this mean. We also quantified WBAS, head, fore-and hind leg onsets to sideslip (+2 m/s). For each body part, we quantified the mean response during the final second of stimulation, and onset was defined as the first time point where the response exceeded 80% of this mean.

### Statistics

Throughout the text *n* refers to individual repetitions, whereas *N* refers to individual neurons (electrophysiology) or animals (behaviour). All data are presented as median and interquartile range, unless otherwise indicated.

Statistical analyses were performed in Prism 10.4.0 (GraphPad Software), except for circular statistics, which were conducted using the CircStat toolbox for MATLAB (*Berens, 2009*). The results of the statistical tests are given in the text, figure legends and in *Tables 1 and 2*. For behavioural data the p-values were Bonferroni adjusted (*Abdi, 2007*) for multiple comparisons (*Table 2*).

## Acknowledgements

We thank Biomedical Engineering at SAHLN and the Botanic Gardens of Adelaide for their ongoing support. We thank the editor and the two Reviewers for their feedback which greatly improved our paper. This research was funded by the US Air Force Office of Scientific Research (AFOSR, FA9550-19-1-0294 and FA9550-23-1-0473) and the Australian Research Council (ARC, DP210100740, DP230100006, DP250100698, and DP250104770).

## Additional information

### Funding

| Funder | Grant reference number | Author |
| --- | --- | --- |
| US Air Force of Scientific Research | FA9550-19-1-0294 | Karin Nordström |
| US Air Force of Scientific Research | FA9550-23-1-0473 | Yuri Ogawa Karin Nordström |
| Australian Research Council | DP210100740 | Karin Nordström |
| Australian Research Council | DP230100006 | Karin Nordström |
| Australian Research Council | DP250104770 | Yuri Ogawa |
| Australian Research Council | DP250100698 | Karin Nordström |

| Funder | Grant reference number | Author |
|--------|------------------------|--------|

The funders had no role in study design, data collection, and interpretation, or the decision to submit the work for publication.

## Author contributions

Sarah Nicholas, Conceptualization, Data curation, Formal analysis, Validation, Investigation, Visualization, Methodology, Writing – original draft, Project administration; Katja Sporar Klinge, Data curation, Formal analysis, Validation, Investigation, Visualization, Methodology, Writing – review and editing; Luke Turnbull, Validation, Investigation, Methodology, Writing – review and editing; Annabel Moran, Aika Young, Investigation, Writing – review and editing; Yuri Ogawa, Data curation, Formal analysis, Supervision, Validation, Investigation, Visualization, Methodology, Writing – original draft, Project administration; Karin Nordström, Conceptualization, Resources, Formal analysis, Supervision, Funding acquisition, Validation, Visualization, Writing – original draft, Project administration

## Author ORCIDs

Sarah Nicholas  https://orcid.org/0000-0002-5555-9421
Katja Sporar Klinge  https://orcid.org/0009-0004-7224-6568
Aika Young  https://orcid.org/0009-0000-5028-6164
Yuri Ogawa  https://orcid.org/0000-0002-9708-7063
Karin Nordström  https://orcid.org/0000-0002-6020-6348

Reviewer #1 (Public review): https://doi.org/10.7554/eLife.109795.3.sa1
Reviewer #2 (Public review): https://doi.org/10.7554/eLife.109795.3.sa2
Author response https://doi.org/10.7554/eLife.109795.3.sa3

# Additional files

## Supplementary files

MDAR checklist

## Data availability

All data and analysis scripts have been submitted to DataDryad: https://doi.org/10.5061/dryad.tb2rbp0fd.

The following dataset was generated:

| Author(s) | Year | Dataset title | Dataset URL | Database and Identifier |
|-----------|------|---------------|-------------|-------------------------|
| Sarah N, Sporar KK, Luke T, Annabel M, Aika Y, Yuri O, Karin N | 2026 | Sexual dimorphism in sensorimotor transformation of insect optic flow | https://doi.org/10.5061/dryad.tb2rbp0fd | Dryad Digital Repository, 10.5061/dryad.tb2rbp0fd |

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
