## [Editor Report · eLife Assessment]

Hoverflies are known for their sexually dimorphic visual systems and exquisite flight behaviors. This **valuable** study reports how two types of visual descending neurons differ between males and females in their motion- and speed-dependent responses, yet surprisingly, the behavior they control lacks any sexual dimorphism. The results **convincingly** support these findings, which will be of interest for studies of visuomotor transformations and network-level brain organization.

---

## [Referee Report · Reviewer #1 (Public review)]

Summary:

Hoverflies are renowned for their striking sexual dimorphism in eye morphology and early visual system physiology, as well as in sexually dimorphic behaviors. Surprisingly, male and female flight behaviors in response to optic flow exhibit only subtle differences. Nicholas et al. investigate the sensorimotor transformation of sexually dimorphic visual information into flight steering commands via descending neurons. Using a combination of intracellular and extracellular recordings, neuroanatomical analysis, and behavioral assays, the authors convincingly demonstrate that descending neurons-particularly at high optic flow velocities-exhibit pronounced sexual dimorphisms, while wing steering responses remain largely monomorphic. The study highlights a very interesting discrepancy between neuronal and behavioral response properties.

More specifically, the authors focused on two types of descending neurons that receive inputs from well-characterized wide-field sensitive tangential cells: OFS DN1 and OFS DN2. Their likely counterparts in Drosophila connect to neck, wing and haltere neuropils. The authors characterized the visual response properties of these two neuronal classes in both male and female hoverflies and identified several interesting differences. They then presented the same set of stimuli, tracked wing beat amplitude and analyzed the sum and the difference of right and left wing beat amplitude as a readout of lift or thrust, and yaw turning, respectively. Behavioral responses showed little to no sexual dimorphism, despite the observed neuronal differences.

Strengths:

I find the question very interesting and the results both convincing and intriguing. A fundamental goal in neuroscience is to link neuronal responses and behavior. The current study highlights that the transformations - even at the level of descending neurons to motoneurons - is complex and less straightforward than one might expect.

Weaknesses:

The authors investigated two types of descending neurons, but it was not clear to me how many other descending neurons are thought to be involved in wing steering responses to wide-field motion. I would suggest providing a more in-depth overview of what is known in hoverflies and Drosophila, since the conclusions drawn from the study would be different if these two types were the only descending neurons involved, as opposed to representing a subset of the neurons conveying visual information to the wing neuropil.

Both neuronal classes have counterparts in Drosophila that also innervate neck motor regions. The authors filled hoverfly DNs in intracellular recordings to characterize their arborization in the ventral nerve cord. In my opinion, these anatomical data could be further exploited and discussed a bit more: is the innervation in hoverflies also consistent with connecting to the neck and haltere motor regions? Are there any obvious differences and similarities to the Drosophila neurons mentioned by the authors? If the arborization also supports a role in neck movements, the authors could discuss whether they would expect any sexual dimorphism in head movements.

Revision comment:

I thank the authors for their detailed replies to my questions and the additional clarifications and analysis included in the paper. All my concerns have been addressed.

---

## [Referee Report · Reviewer #2 (Public review)]

Summary:

Many fly species exhibit male-specific visual behaviors during courtship while little is known about the circuit underlying the dimorphic visuomotor transformations. Nicholas et al focus on two types of visual descending neurons (DNs) in hoverflies, a species in which only males exhibit high-speed pursuit of conspecifics. They combined electrophysiology and behavior analysis to identify these DNs and characterize their response to a variety of visual stimuli in both male and female flies. The results show that the neurons in both sexes have similar receptive fields but exhibit speed-dependent dimorphic responses to different optic flow stimuli.

Strengths:

Hoverflies, though not a common model system, show very interesting dimorphic behaviors and provide a unique and valuable entry point to explore the brain organization behind sexual dimorphism. The findings here are not only interesting on their own right but will also likely inspire those working in other systems, particularly Drosophila.

The authors employed rigorous morphology, electrophysiology, and behavior methods to deliver comprehensive characterization of the neurons in question. The precision of the measurements allowed for identifying a subtle and nuanced neuronal dimorphism and set a standard for future work in this area.

Weaknesses:

I'd like to thank the authors for the revised manuscript, especially the new analyses and figures. Most of my earlier concerns have been satisfactorily addressed by now. Interested readers are kindly referred to the authors' responses for the discussion of the limitations of this work.

---

## [Author Response]

The following is the authors’ response to the original reviews.

**eLife Assessment**
Hoverflies are known for their sexually dimorphic visual systems and exquisite flight behaviors. This valuable study reports how two types of visual descending neurons differ between males and females in their motion- and speed-dependent responses, yet surprisingly, the behavior they control lacks any sexual dimorphism. The results convincingly support these findings, which will be of interest for studies of visuomotor transformations and network-level brain organization.

This statement perfectly recapitulates our findings.

**Public Reviews:**

**Reviewer #1 (Public review):**
Summary:Hoverflies are known for a striking sexual dimorphism in eye morphology and early visual system physiology. Surprisingly, the male and female flight behaviors show only subtle differences. Nicholas et al. investigate the sensori-motor transformation of sexually dimorphic visual information to flight steering commands via descending neurons. The authors combined intra- and extracellular recordings, neuroanatomy, and behavioral analysis. They convincingly demonstrate that descending neurons show sexual dimorphisms - in particular at high optic flow velocities - while wing steering responses seem relatively monomorphic. The study highlights a very interesting discrepancy between neuronal and behavioral response properties.

Thank you for this summary. Most of the statement perfectly recapitulates the main findings of our paper. However, we want to emphasize that some hoverfly flight behaviors are strongly sexually dimorphic, especially those related to courtship and mating. Indeed, only male hoverflies pursue targets at high speed, chase away territorial intruders, and pursue females for mating. However, other flight behaviours, such as those related to optomotor responses and flights between flowers when feeding, are not sexually dimorphic. We have amended the Introduction and Discussion to make the difference between flight behaviors more clear. Please see lines 77 and 305 onwards.

More specifically, the authors focused on two types of descending neurons that receive inputs from well-characterized wide-field sensitive tangential cells: OFS DN1, which receives inputs from so-called HS cells, and OFS DN2, which receives input from a set of VS cells. Their likely counterparts in Drosophila connect to the neck, wing, and haltere neuropils. The authors characterized the visual response properties of these two neuronal classes in both male and female hoverflies and identified several interesting differences. They then presented the same set of stimuli, tracked wing beat amplitude, and analyzed the sum and the difference of right and left wing beat amplitude as a readout of lift or thrust, and yaw turning, respectively. Behavioral responses showed little to no sexual dimorphism, despite the observed neuronal differences.

Thank you for this very nice summary of our work. We want to clarify that LPTC input to DN1 and DN2 has not been shown directly in hoverflies using e.g. dye coupling, or dual recordings. Instead, the presumed HS and VS input is inferred from morphological and physiological DN evidence, and comparisons to similar data in *Drosophila* and blowflies. We have amended the Introduction to clarify this. Please see line 64 onwards. The rest of the paragraph perfectly recapitulates the main findings of our paper.

Strengths:I find the question very interesting and the results both convincing and intriguing. A fundamental goal in neuroscience is to link neuronal responses and behavior. The current study highlights that the transformations - even at the level of descending neurons to motoneurons - are complex and less straightforward than one might expect.

Thank you.

Weaknesses:The authors investigated two types of descending neurons, but it was not clear to me how many other descending neurons are thought to be involved in wing steering responses to wide-field motion. I would suggest providing a more in-depth overview of what is known about hoverflies and Drosophila, since the conclusions drawn from the study would be different if these two types were the only descending neurons involved, as opposed to representing a subset of the neurons conveying visual information to the wing neuropil.

This is a great point. There are around 1000 fly descending neurons identified in *Drosophila*, of which many could respond to widefield motion, without being specifically tuned to widefield motion. In *Drosophila*, at least 35 descending neuron types receive input in the part of the brain where the LPTC outputs are located, and at least 29 descending neuron types project to the wing motor neuropil. Thus, it is more than likely that other neurons project visual widefield motion information to the wing neuropil. Furthermore, we only measured wing beat amplitude (WBA) as seen in the horizontal plane, as we were filming from above. As such, other wing angle changes and rotations are not quantified. We have amended our Introduction (see line 53 onwards) and Discussion (see line 320 onwards) to address these important points.

Both neuronal classes have counterparts in Drosophila that also innervate neck motor regions. The authors filled the hoverfly DNs in intracellular recordings to characterize their arborization in the ventral nerve cord. In my opinion, these anatomical data could be further exploited and discussed a bit more: is the innervation in hoverflies also consistent with connecting to the neck and haltere motor regions? Are there any obvious differences and similarities to the Drosophila neurons mentioned by the authors? If the arborization also supports a role in neck movements, the authors could discuss whether they would expect any sexual dimorphism in head movements.

These are all great points. We did not see any clear arborizations to the frontal nerve (FN), where we would expect to find the neck motor neurons (NMNs). In addition, while we did see fine arborizations throughout the length of the thoracic ganglion, we saw no strong outputs projecting directly to the haltere nerve (HN). In the revised version of the MS we have modified figure 4 (morphological characterization) to show a magnification of the thoracic ganglion to clarify this.

There are important differences between the morphology of DN1 and DN2 in hoverflies and DNHS1 and DNOVS2 in *Drosophila*, in terms of their projections in the thoracic ganglion. For example, In *Drosophila* DNOVS2, there are several fine branches along the length of the neuron in the thoracic ganglia. Similarly, we found fine branches in *Eristalis tenax* DN2, however, in addition, we found a wide branch projecting to the area of the thoracic ganglion where the prothoracic and pterothoracic nerves likely get their inputs, which we also found in *Eristalis tenax* OFS DN1 (Figure 4). This suggests that both neurons could contribute to controlling the wings and/or the forelegs (which is why we quantified the WBA). In *Drosophila* DNOVS1, there is a similar fat branch to the prothoracic and pterothoracic nerves, Furthermore, while *Drosophila* DNHS1 and DNOVS2 have different morphology, DN1 and DN2 in *Eristalis* looked similar. We have modified the Results section to make this clear, see line 193 onwards.

In addition, to investigate this further, our revised version of the MS includes analysis of the movement of different body parts (the head angle, fore- and hindleg extension) to investigate this further, and to look for sexual dimorphism. Unfortunately, however, this did not include the halteres, as they cannot be seen well in the videos. The new data can be seen in Figure 7.

**Reviewer #2 (Public review):**
Summary:Many fly species exhibit male-specific visual behaviors during courtship, while little is known about the circuit underlying the dimorphic visuomotor transformations. Nicholas et al focus on two types of visual descending neurons (DNs) in hoverflies, a species in which only males exhibit high-speed pursuit of conspecifics. They combined electrophysiology and behavior analysis to identify these DNs and characterize their response to a variety of visual stimuli in both male and female flies. The results show that the neurons in both sexes have similar receptive fields but exhibit speed-dependent dimorphic responses to different optic flow stimuli.

This statement perfectly recapitulates the main findings of our paper. As mentioned above, while hoverfly flight behaviors related to courtship and mating are strongly sexually dimorphic, other flight behaviours, such as those related to optomotor responses and flights between flowers when feeding, are not. We have amended the Introduction and Discussion to make the difference between flight behaviors more clear. Please see lines 77 and 305 onwards.

Strengths:Hoverflies, though not a common model system, show very interesting dimorphic behaviors and provide a unique and valuable entry point to explore the brain organization behind sexual dimorphism. The findings here are not only interesting on their own right but will also likely inspire those working in other systems, particularly Drosophila.

Thank you.

The authors employed rigorous morphology, electrophysiology, and behavior methods to deliver a comprehensive characterization of the neurons in question. The precision of the measurements allowed for identifying a subtle and nuanced neuronal dimorphism and set a standard for future work in this area.

Thank you.

Weaknesses:Cell-typing using receptive field preferred directions (RFPDs): if I understood correctly, this classification method mostly relies on the LPDs near the center of the receptive field (median within the contour in Fig.1). I have two concerns here. First, this method is great if we are certain there are only two types of visual DNs as described in the manuscript. But how certain is this? Given the importance of vision in flight control, I would expect many DNs that transmit optic flow information to the motor center. I'd also like to point out that there are other lobula plate tangential cells (LPTCs) than HS and VS cells, which are much less studied and could potentially contribute to dimorphic behaviors.

This is very true, and important. As mentioned above, in *Drosophila* there are 35 descending neuron types with inputs on the dorsal surface of the brain (labelled DNp1-35), suggesting that they could receive input from LPTCs. However, only 3 of these have been shown physiologically and morphologically to receive LPTC input, in blowflies and *Drosophila* (DNHS1, DNOVS1, DNOVS2). Note that in both blowflies and fruitflies DNOVS1 gives graded responses, and no action potentials, meaning that we would not be able to record from it using extracellular electrophysiology.

We previously used clustering techniques to show that in *Eristalis*, we can reliably distinguish two types of optic flow sensitive DNs from extracellular electrophysiological data, based on a range of receptive field parameters, and we think that these correspond to DNHS1 and DNOVS2 in *Drosophila* (Nicholas et al, J Comp Physiol A, 2020, cited in paper). As mentioned above in response to Reviewer 1, this does not mean that there are no other neurons that could respond to widefield optic flow, and which might be involved in the WBA we recorded in the paper. However, the point of this paper was not to conclusively show that there are only two optic flow sensitive descending neurons. The point was to say that there are two quite distinct optic flow sensitive neurons that have similar receptive fields in males and females, while their velocity response functions differ between males and females.

We have modified the Introduction (see lines 53 and 64 onwards) and Discussion to make these important points clear to the Reader, including a mention of the 45-60 LPTCs that exist in the lobula plate, and what their role might be.

Second, this method feels somewhat impoverished given the richness of the data. The authors have nicely mapped out the directional tuning for almost the entire visual field. Instead of reducing this measurement to 2 values (center and direction), I was wondering if there is a better method to fully utilize the data at hand to get a better characterization of these DNs. As the authors are aware, local features alone can be ambiguous in characterizing optic flows. What's more, taking into account more global features can be useful for discovering potentially new cell types.

This is a great point, and we did analyse other receptive field properties in this study (shown in previous supp fig 1). In addition, and as mentioned above, we have published a clustering analysis across receptive field properties of these neurons (Nicholas et al, J Comp Physiol A, 2020, cited in paper). The point that we attempted to make in this paper was that by using two strikingly simple metrics, we can reliably distinguish which of the two neuron types we are recording from simply based on azimuthal location and overall directional preference. This makes automated analysis very straightforward. Indeed, we now use this routinely to ID what neuron we are recording from computationally, rather than making a human-based assumption.

However, we agree that this needs to be shown, and that further in depth analysis was warranted. Therefore, we have provided additional receptive field analysis and clustering (see new supplementary figure 1) and associated text. We also want to highlight that all data is uploaded to Data Dryad for anyone interested in doing additional in-depth analyses.

Line 131, it wasn't clear to me why full-screen stimuli were used for comparison here, instead of the full receptive field maps. Male flies exhibit sexual dimorphic behaviors only during courtship, which would suggest that small-sized visual stimuli (mimicking an intruder or female conspecific) would be better suited to elicit dimorphic neuronal responses. A similar comment applies to the later results as well. Based on the receptive field mapping in Figure 1, I'm under the impression that these 2 DN types are more suited to detect wide-field optic flows, those induced by self-motion as mentioned in the manuscript. The results are still very interesting, but it's good to make this point clear early on to help set appropriate expectations. Conversely, this would also suggest that there are other visual DN types that are responsible for the courtship-related sexually dimorphic behaviors.

Thank you for mentioning these important points. Our reasoning for using full-screen stimuli for the analysis on line 131 was that since we used the small sinusoidal gratings for mapping the receptive fields, and to subsequently classify the neurons, it would be unfair to use the same data to investigate potential sexual dimorphism. I.e., we selected neurons that fulfilled certain criteria, and then we cannot rightfully use the same criteria to determine differences. This was not explicitly mentioned in the paper, so we have modified the text to make this clear to the Reader, see lines 142 onwards.

However, in Supp Figure 2d/e we show that there are no striking receptive field differences between males and females in terms of receptive field center nor directional preference. In Supp Figure 2f we also show that there is no difference between male and female receptive field height and width. We have modified the text to draw the Reader’s attention to this figure, and also mention the additional analysis done in response to the comment above.

As a side note, I personally expected at least DN1 to have a smaller receptive field in males, as the hoverfly HSN is strikingly sexually dimorphic (Nordström et al, Curr Biol 2008). However, while optic flow sensitive DNs do respond to small objects (see e.g. the J Comp Physiol paper mentioned above) we did not detect any obvious sexual dimorphism in receptive field properties. Indeed, we think that a different subset of DNs control parts of target pursuit behavior (target selective DNs (TSDNs)). This is now addressed in the modified version of the paper, see line 89-92.

**Recommendations for the authors:**

**Reviewer #1 (Recommendations for the authors):**
(1) I think that the additional measurement of head turns in response to some of the stimuli that showed the strongest sexual dimorphism would be very interesting, but I fully acknowledge that this might be beyond the scope of the current paper or technically too challenging, requiring additional cameras and a whole new tracking software, etc.

We have added an additional figure to the paper, with associated text, showing the response of the head, fore- and hindlegs to the same stimuli, as far as we could extract them with only one camera filming from above. The new data can be found in the new figure 7, and associated text.

(2) Are the onset measurements for WBD comparable across flight manoeuvres, given that they are limited to a single projection plane?

This is a great point, and we have now added this caveat in the text, see line 261-262.

(3) Line 62 - typo: DNp15 not NDp15.

Thank you, fixed.

**Reviewer #2 (Recommendations for the authors):**
(1) Related to a comment earlier, in the Introduction, it is mentioned that there are 3 optic flow-sensitive DNs in Drosophila and blowfly. However, I don't see convincing evidence for this in the cited references, none of which have exclusively surveyed all the DNs.

We have revised this to say that 3 neuron have been identified morphologically and physiologically, but that does not mean that there are no others. Please see line 60 onwards.

(2) Line 142 and Supplementary Figure 3, this is stated in the next section, but I think it's better to make it clear that DN2 in females has a higher spontaneous rate before mentioning the starfield. Please also specify if the stationary starfield affects the DN2 rate at all in the female flies.

Great points. We now describe the spontaneous rate before mentioning the responses to moving starfield stimuli, and highlight that there is no difference between no stimulus (pre-stimulation) and a stationary stimulus. Please see lines 155 onwards.

(3) Line 34, 'redress' should be 'to address'.

Thank you, fixed.

(4) Line 59, a bit unclear to me what this sentence is trying to say. Also, I wouldn't say LPTCs are 'indirect' in the sensorimotor transformation -- it's a necessary link in this pathway, no?

That was indeed a strange sentence. We have simplified it to the following: “LPTCs project to the inferior posterior slope[6], where they synapse with descending neurons[7,8]. In *Drosophila* at least 35 descending neuron types have their inputs in the posterior surface of the brain (named DNp1-35) [9].”

(5) Figures:This is a formatting problem. The figure legends are separated from the figures, and there are no titles on the figures to indicate which one is which.

We are sorry about this. We have added labels to the figures.

Figure 1: What kind of geographic projections are these? The azimuth axis is not labeled.

These stimuli were not perspective corrected, and therefore the RF maps simply reflect the visual monitor. We have clarified this in the figure legend, including mentioning that the axis label is the same for elevation and azimuth.

Figure 2a: The error bars are not aligned to the angular axis.

These have now been aligned.

Supplement Figure 2b: I'm not sure why there are two measurements at each stimulus orientation. The bottom panel is confusing -- what do you mean by 'receptive field location'? And what does this red arrow/line mean in the bottom panel?

Thank you for pointing this out. The figure was supposed to help the reader understand our transformations, so it’s great to know that it needed further explanation. To address this, we have added extra text and panel labels, please see lines 520 onwards.

(6) Methods:Line 356: Maybe a picture or schematic drawing would be helpful to explain the setup. For instance, it's unclear what 32 degrees here refers to.

This is a great suggestion, and a pictogram explaining the set-up can now be seen in Supplementary Fig. 6b.

Line 404: What does it mean that 'spatially interpolate 10 times'?

This sentence has been changed to “After subtracting the spontaneous rate, calculated for 0.8 s preceding stimulus onset (dotted line, inset, Fig. 1b, e), we interpolated the resulting local maximum responses to a ten-fold higher spatial resolution (colour coding, Fig. 1a, d).”

Line 405: How to determine the center from the 50% contour?

We have modified the Methods to explain how this was done, please see lines 478 onwards.

Line 408: Please explain more explicitly how LPD and LMS are computed.

We have modified the Methods to explain how this was done, please see lines 488 onwards.

Line 418: Is reference 42 correct? I could be wrong, but this reference seems to be talking about target-selective DNs rather than optic flow-sensitive DNs?

Yes, this reference is correct. In a supp figure to ref 42, we show data from optic flow sensitive neurons, but not their receptive fields. Thanks for checking.

Line 426: Are the full-screen stimuli presented in 8 directions too? Do I understand correctly that the preferred direction vector for the full-screen stimuli is extracted from a cosine fit, which is slightly different from the 'receptive field preferred direction' in the receptive field mapping measurement, which is the median of all the 'local preferred direction' (which are from the cosine fit)?

We have modified the text to make this clear, please see lines 519 onwards, as well as the receptive field analysis, please see lines 474 onwards.